# Discovery of plant chemical defence mediated by a two-component system involving *β*-glucosidase in *Panax* species

Li-Juan Ma[1,9], Xiao Liu [2,9], Liwei Guo[3], Yuan Luo[2], Beibei Zhang[2,4], Xiaoxue Cui[2], Kuan Yang[3], Jing Cai [5], Fang Liu[1], Ni Ma[6], Feng-Qing Yang [7], Xiahong He[3,8] ✉, She-Po Shi [2] ✉ & Jian-Bo Wan [1] ✉

Plants usually produce defence metabolites in non-active forms to minimize the risk of harm to themselves and spatiotemporally activate these defence metabolites upon pathogen attack. This so-called two-component system plays a decisive role in the chemical defence of various plants. Here, we discovered that *Panax notoginseng*, a valuable medicinal plant, has evolved a two-component chemical defence system composed of a chloroplast-localized *β*-glucosidase, denominated PnGH1, and its substrates 20(*S*)-protopanaxadiol ginsenosides. The *β*-glucosidase and its substrates are spatially separated in cells under physiological conditions, and ginsenoside hydrolysis is therefore activated only upon chloroplast disruption, which is caused by the induced exoenzymes of pathogenic fungi upon exposure to plant leaves. This activation of PnGH1-mediated hydrolysis results in the production of a series of less-polar ginsenosides by selective hydrolysis of an outer glucose at the C-3 site, with a broader spectrum and more potent antifungal activity in vitro and in vivo than the precursor molecules. Furthermore, such *β*-glucosidase-mediated hydrolysis upon fungal infection was also found in the congeneric species *P. quinquefolium* and *P. ginseng*. Our findings reveal a two-component chemical defence system in *Panax* species and offer insights for developing botanical pesticides for disease management in *Panax* species.

In nature, plants deploy an enormous array of secondary metabolites to defend against pathogens and herbivores[1]. These defensive metabolites can be classified into two broad categories: constitutive defence metabolites (phytoanticipins), which act directly on pathogens or herbivores, and inducible defence compounds (phytoalexins), which are specifically produced in response to aggressors[2]. Compared with constitutive defence, inducible defence is more energetically economical and plays an essential role in plant chemical defence[3]. The

[1]State Key Laboratory of Quality Research in Chinese Medicine, Institute of Chinese Medical Sciences, University of Macau, Macao, China. [2]Modern Research Center for Traditional Chinese Medicine, Beijing University of Chinese Medicine, Beijing, China. [3]State Key Laboratory for Conservation and Utilization of Bio-Resources in Yunnan, Yunnan Agricultural University, Kunming, Yunnan, China. [4]State Key Laboratory of Bioactive Substance and Function of Natural Medicines, Institute of Materia Medica, Chinese Academy of Medical Sciences and Peking Union Medical College, 100050 Beijing, China. [5]School of Ecology and Environment, Northwestern Polytechnical University, Xi'an, Shaanxi, China. [6]Department of Product Development, Wenshan Sanqi Institute of Science and Technology, Wenshan University, Wenshan, Yunnan, China. [7]Department of Pharmaceutical Engineering, School of Chemistry and Chemical Engineering, Chongqing University, 401331 Chongqing, China. [8]Ministry of Education Key Laboratory for Forest Resources Conservation and Utilization in the Southwest Mountains of China, Southwest Forestry University, 650224 Kunming, Yunnan, China. [9]These authors contributed equally: Li-Juan Ma, Xiao Liu. ✉e-mail: hxh@swfu.edu.cn; shishepo@163.com; jbwan@um.edu.mo

specialized metabolites in inducible defence are usually toxic to the plant, pathogens, or herbivores. Accordingly, plants have evolved two-component chemical defence systems that prevent autotoxicity to help balance plant growth and defence needs[1]. For instance, defence compounds are often stored in a non-active glycosylated form[1,4]. In addition to protecting plants from autotoxicity, glycosylation increases the solubility of defence compounds, thereby facilitating their storage, typically in vacuoles[5]. Upon plant cell damage caused by herbivory or pathogen invasion, glycosylated compounds can be selectively hydrolyzed at glycosidic linkages, yielding toxic defence compounds[6]. This hydrolysis reaction is catalyzed by $\beta$-glycosidase, which is spatially separated from its substrates under physiological conditions but encounters them upon tissue damage. Thus, this two-component chemical defence system comprises glycosylated defence compounds and corresponding $\beta$-glycosidases and provides an immediate chemical response against herbivores or pathogens[7]. Although $\beta$-glycosidases have been shown to play a decisive role in the chemical defence of various plants, these enzymes have not yet been reported in *Panax* species.

The genus *Panax* in the taxonomic family Araliaceae encompasses several important ginseng species with high medicinal and economic value, including *Panax notoginseng* (Burk.) F.H. Chen (Notoginseng), *Panax ginseng* C.A. Mey (Asian Ginseng) and *Panax quinquefolium* L. (American Ginseng)[8,9]. The cultivation of *P. notoginseng* is primarily restricted to southern China; the roots of this species are highly valued for their use in the prevention and treatment of diverse hematological diseases and ischemic cardiovascular diseases[10–12]. *P. notoginseng* is highly sensitive to sunlight and requires shady, warm, damp environments for growth[13]. These stringent requirements for cultivation and long harvest cycles increase the vulnerability of the plant to pathogen infection. Over 20 foliar and root diseases that affect *P. notoginseng* growth have been identified, including round spots caused by *Mycocentrospora acerina*, black spots caused by *Alternaria panax*, and rotten roots caused by *Fusarium oxysporum*. These fungal diseases greatly threaten the quality and production of *P. notoginseng* and severely hinder the sustainable development of its industry. During evolution, *P. notoginseng* deploys an enormous array of specialized secondary metabolites that help them solve pathogen attacks. Dammarane triterpenoid saponins with 20(*S*)-protopanaxadiol (PPD) and 20(*S*)-protopanaxatriol (PPT) aglycon moieties, referred to as ginsenosides, perform defensive functions in plants[14] and are also the main pharmacological components of *P. notoginseng*[15]. These ginsenosides are distributed in various parts of *P. notoginseng* plants, such as rhizomes, roots, leaves, inflorescences, and infructescences[16,17]. The ginsenosides Rc, Rb2, and Rb3, classified as the PPD type ginsenosides, are the main chemical components of *P. notoginseng* leaves (PNLs) but are present at trace levels in the roots[16,17]. Our previous studies have shown that the main ginsenosides in PNLs, which have a common $\beta$-1,2-glucosidic linkage at the C-3 site, can be entirely hydrolyzed to form corresponding rare ginsenosides by selectively removing one outer glucose residue during ultrasonic extraction in water, and this ginsenoside hydrolysis has been speculatively attributed to a ginsenoside-hydrolyzing glucosidase present in PNLs[10,18]. However, the $\beta$-glucosidase responsible for this reaction and its role in plant chemical defence is largely unknown.

Herein, we provide evidence demonstrating the existence of a two-component chemical defence system in *P. notoginseng*. PnGH1, a $\beta$-glucosidase localized to the chloroplast, was discovered to specifically hydrolyze PPD ginsenosides in PNLs by selectively removing one outer glucose residue at the C-3 site. Due to spatial separation between $\beta$-glucosidase and its substrates in cells, ginsenoside hydrolysis was triggered only upon chloroplast disruption caused by the induced exoenzymes of pathogens. Compared to the non-hydrolyzed substrates, the hydrolysis products exhibited more potent inhibition of pathogen growth both in vitro and in vivo. Similar $\beta$-glucosidase-mediated ginsenoside hydrolysis was found in congeneric species upon fungal infection, including in *P. ginseng* and *P. quinquefolium*.

## Results

### Localized ginsenoside hydrolysis occurs in the lesion area of *P. notoginseng* leaves with round spot disease in open field

Round spot caused by the necrotrophic fungus *M. acerina* is a primary foliar disease occurring in cultured *P. notoginseng*. Given its high prevalence and severe consequences, we investigated chemical changes of specialized metabolites in PNLs with round spot disease collected from the open field. Specifically, different zones (illustrated in Fig. 1a) of infected leaves from a single *P. notoginseng* plant were harvested for chemical analysis by LC-UV, including analysis of the lesion area (Zone I), a healthy area of the same leaf (Zone II), and a healthy area of a neighboring leaf (Zone III). Surprisingly, the chemical profile of the lesion tissue was clearly different from those of the other two zones and healthy leaves from a non-infected plant (CON) (Fig. 1b). The main components identified in healthy leaves and lesion tissue were designated compounds **1**–**4** and **1a**–**4a**, respectively, by the aid of their reference standards and LC−MS analysis. The most plausible explanation for these differences is that ginsenoside hydrolysis might occur locally in the lesion tissue of PNLs with round spot disease. Specifically, ginsenosides Rb1 (**1**), Rc (**2**), Rb2 (**3**) and Rb3 (**4**), the main components of PNLs, were hydrolyzed to gypenoside XVII (**1a**), notoginsenoside Fe (**2a**), ginsenoside Rd2 (**3a**), and notoginsenoside Fd (**4a**), respectively, via the selective hydrolysis of an outer glucose from the $\beta$ (1→2)-glycosidic linkage at C-3 without cleavage of other glycosidic linkages (Fig. 1c). In contrast, Fc (**5**) conjugated with a Glu-Glu-Xyl carbohydrate chain at C-3 (Supplementary Fig. 1) was not hydrolyzed. Furthermore, to quantitatively characterize the hydrolysis process in PNLs under various stresses, the ratios of Fd to Rb3 + Fd and Fe to Rc + Fe were defined as the hydrolysis turnover. Rb3 (**4**) and Rc (**2**) were the most abundant and second most abundant ginsenosides in PNLs, and their hydrolysis products were Fd (**4a**) and Fe (**2a**), respectively. As a result, the hydrolysis turnovers in Zone I of PNLs with round spot disease were significantly higher than those in other zones and the leaves of noninfected plants (Fig. 1d).

To examine whether the observed ginsenoside hydrolysis was associated with a stress type-specific response, leaf tissues of *P. notoginseng* plants exposed to various biotic and abiotic stresses were collected in the open field. The stresses to which these plants were subjected included sunscald (a physiological disorder caused by exposure to excess solar radiation), root-knot nematode disease (a pathogenic root disease caused by *M. hapla* Chitwood), and round spot disease. A significant increase in ginsenoside hydrolysis was observed only in PNLs with round spot disease, regardless of the type of abiotic stress in open field conditions (Fig. 1e). These results indicate that ginsenoside hydrolysis is locally triggered in the lesion area of PNLs with round spot disease in the open field.

### Ginsenoside hydrolysis is observed exclusively in lesion tissue of PNLs infected with foliar pathogenic fungi in the greenhouse

As open-field conditions are highly complex, a greenhouse experiment was conducted to verify the open-field findings. Fungal discs were punched from the edge of a colony of fungus actively growing on PDA plates, and one-year-old *P. notoginseng* were inoculated by placing the fungal disc on the leaf surface (Fig. 2a). Hydrolysis turnovers in the three zones defined above were then compared at 5 days post-inoculation (DPI). The ratio of Fd/Rb3 + Fd or Fe/Rc + Fe in lesion tissue (Zone I) was apparently increased compared with that in other zones and CON leaves (Fig. 2b), consistent with the results from the open-field study.

To further clarify the correlation between ginsenoside hydrolysis and pathological stress, PNLs were inoculated with different pathogenic and non-pathogenic fungi. Inoculation with *A. panax*, a black

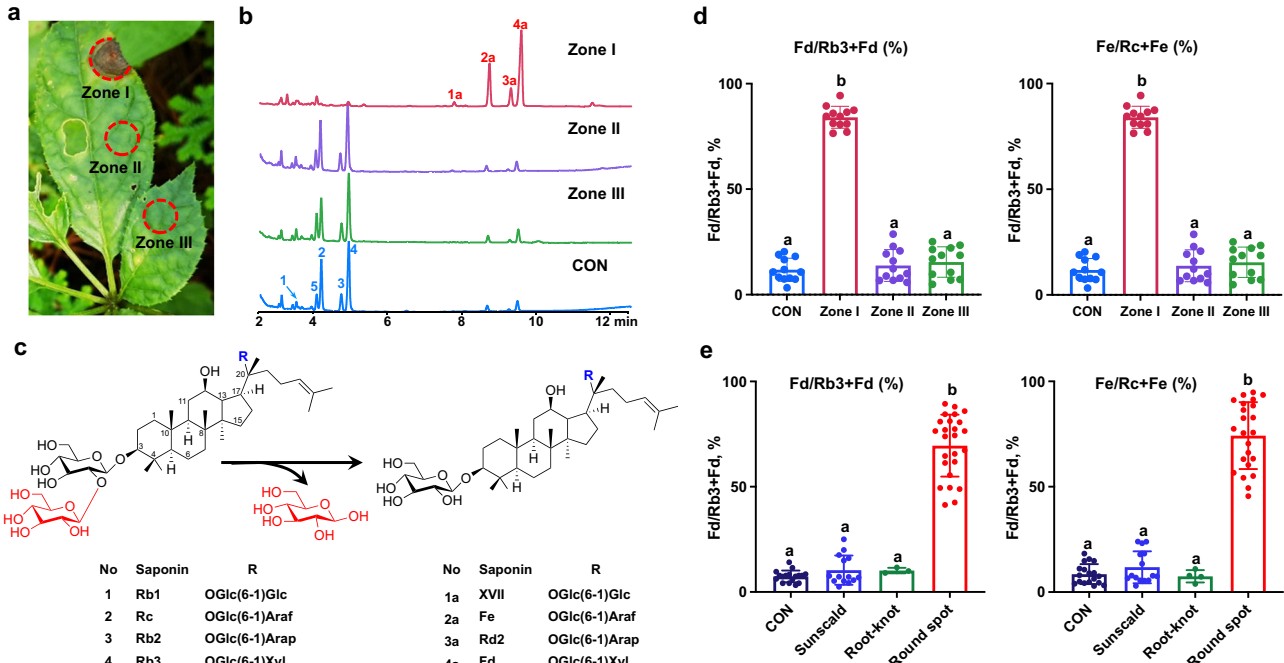

**Fig. 1 | Localized ginsenoside hydrolysis occurs in the lesion area of *P. notoginseng* leaves with round spot disease in open field. a** Different zone of the infected PNL was harvested for chemical analysis. Zone I: lesion area, Zone II: healthy area in the same leaf, Zone III: healthy area in the neighboring leaf. **b** HPLC-UV chromatograms of methanolic extracts of healthy leaf from the non-infected plants (CON) and PNL with round spot disease at different zones collected in the open field, the peak number of ginsenosides represent in the same manner in (**d**). **c** Chemical structures of hydrolysis-involved ginsenosides in PNL and their corresponding products. Glc, $\beta$-D-glucopyranosyl, Araf, α-L-arabinofuranosyl; Arap, α-L-arabinopyranosyl; xyl, $\beta$-D-xylopyranosyl. **d** Hydrolysis turnovers of different zones in PNL with round spot disease, calculated as the ratio of Fd to Fd + Rb3 (left) and Fe to Rc+Fe (right), CON, the healthy leaf from the non-infected plants. $n = 12$ biologically independent samples. **e** Hydrolysis turnovers of corresponding leaf tissue from *P. notoginseng* plants upon different biotic and abiotic stresses, including sunscald (a physiological disorder caused by exposure to excess solar radiation), root-knot nematode disease (a pathogenic root disease caused by *M. hapla* Chitwood) and round spot disease (a pathogenic foliar disease caused by *M. acerina*). $n = 17, 14, 3,$ and 26 biologically independent samples in control, sunscald, root-knot, and round spot groups, respectively. **d, e** The data are expressed as mean ± SD. Statistical significance was calculated using a one-way ANOVA, followed by Tukey's post hoc test. Data with different letters above the bars are significantly different ($p < 0.05$). Source data are provided as a Source Data file.

spot disease pathogen causing foliar necrosis in PNLs, resulted in the development of severe lesions at 5 DPI. Significantly higher hydrolysis turnovers were observed in decayed tissue, compared with that in other zones and CON leaves (Fig. 2c). However, non-pathogenic fungi, such as *F. oxysporum* (a root rot disease pathogen, Fig. 2d) and *Colletotrichum orchidophilum*[19] (an anthracnose disease pathogen found on *Bletilla striata* leaves, Fig. 2e), failed to either infect the leaves or trigger ginsenoside hydrolysis. Our data indicates that ginsenoside hydrolysis is exclusively activated in lesion tissues of PNLs infected by a foliar pathogenic fungus.

To determine whether a broth culture of a pathogenic fungus could cause hydrolysis, Rb3 was spiked into an *M. acerina* culture, and the spiked culture was incubated for 11 days. The hydrolysis product Fd was not detected, and no significant difference in Rb3 content was observed at different intervals (Fig. 2f), indicating that enzymes secreted by *M. acerina* cannot trigger Rb3 hydrolysis.

Next, dynamic chemical changes during fungal infection were characterized in *M. acerina*- and mock-inoculated PNLs in a time-course analysis. At the initial infection stage (0–3 DPI), there was no significant difference in Rb3 and Fd levels in infected tissue (Zone I) between mock-inoculated and *M. acerina*-inoculated plants. At 5 DPI, round or nearly round water-soaked spots were clearly observed in the PNLs (Fig. 2g). Additionally, the level of Rb3 in Zone I was significantly decreased, and its hydrolysis product, Fd, was generated accordingly (Fig. 2h), indicating that ginsenoside hydrolysis was activated upon successful fungal colonization. Interestingly, the levels of the hydrolysis products peaked in infected PNLs at 11 DPI, and the molar amounts of Fd, Fe, Rd2 and XVII in *M. acerina*-infected

PNLs were increased approximately 7.8-fold, 3.2-fold, 2.9-fold, and 2.2-fold, respectively, compared with the amounts of the intact ginsenosides in mock-treated PNLs (Fig. 2h and Supplementary Fig. 2). This excessive accumulation of hydrolysis products in decayed tissue suggests that ginsenoside biosynthesis might also be enhanced, in addition to ginsenoside hydrolysis, to produce greater amounts of hydrolysis products upon fungal infection. Furthermore, no significant difference in Rb3 or Fd levels was observed in either Zone II or Zone III between mock-treated and fungus-infected PNLs during the entire infection period (Fig. 2i, j), confirming that ginsenoside hydrolysis was locally activated in the lesion tissue of PNLs infected with the pathogenic fungus. Similar dynamic changes in other pairs of ginsenosides, including Rc/Fe (Supplementary Fig. 2a), Rb2/Rd2 (Supplementary Fig. 2b), and Rb1/XVII (Supplementary Fig. 2c) were also observed. Still, no significant difference in Fc levels in decayed tissue was found between mock-treated and fungus-infected PNLs (Supplementary Fig. 2d).

## PnGH1, a $\beta$-glucosidase, is responsible for ginsenoside hydrolysis in *P. notoginseng* leaves

To identify the glucosidase potentially involved in regioselective ginsenoside hydrolysis in PNLs, the crude protein obtained from the PNL homogenate was fractionated following an activity-guided protein-purification procedure comprising ammonium sulfate fractionation, ion exchange chromatography and size exclusion chromatography (Supplementary Fig. 3a). This led to the successful purification of the expected glucosidase that could efficiently hydrolyze Rb3 (**4**) to Fd (**4a**). The purified protein was then digested with sequencing-grade

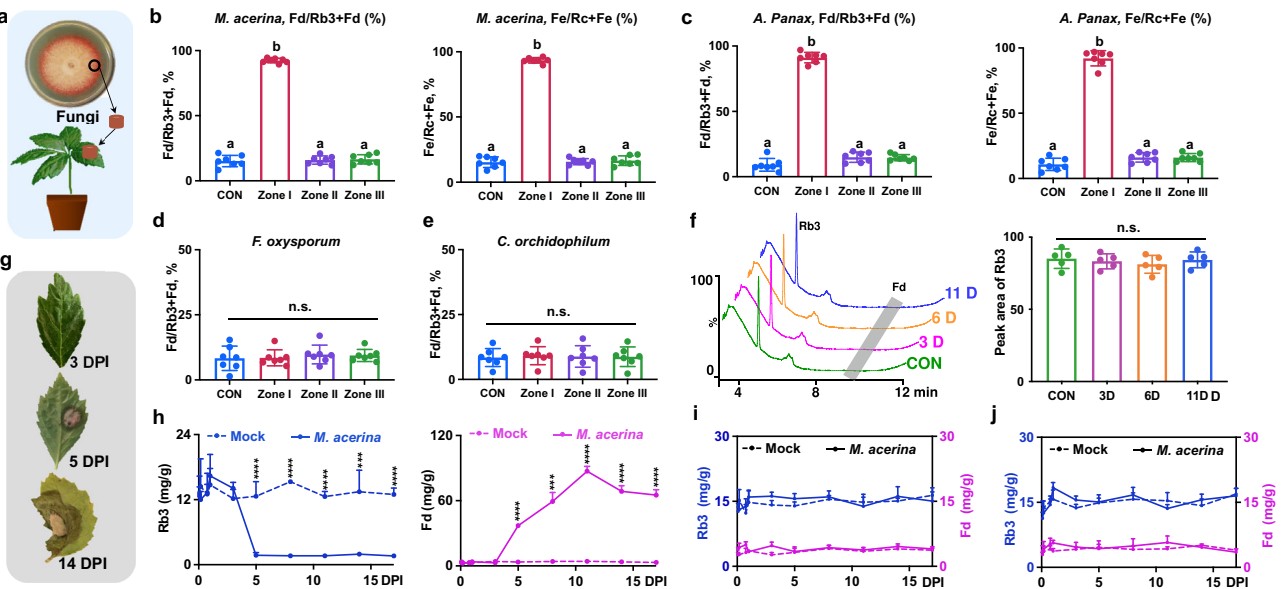

**Fig. 2 | Ginsenoside hydrolysis in lesion tissue of PNL is associated with the infection of foliar pathogenic fungi in the greenhouse. a** Diagram of plant inoculation, a fungal disc (5 mm diameter) was punched from the actively growing edge of the PDA plates, one-year-old *P. notoginseng* plants were inoculated by placing the fungal disc on the surface of leaves. **b–e** Hydrolysis turnovers of different zones in PNL at 5 DPI with foliar pathogenic fungus, *M. acerina* (**b**) and *A. Panax* (**c**), and non-pathogenic fungus, *F. oxysporum* (**d**) and *C. orchidophilum* (**e**). *n* = 7 biologically independent samples for each group. The data are expressed as mean ± SD. Statistical significance was calculated using a one-way ANOVA, followed Tukey's post hoc test. Data with different letters above the bars are significantly different (*p* < 0.05). n.s. not significant. f Broth culture of *M. acerina* fails to activate ginsenoside hydrolysis. HPLC-UV profiles (left) and Rb3 levels in potato dextrose broth (PDB) medium that incubated *M. acerina* with Rb3 at different time points. Rb3 was added to 50 mL of PDB culture of *M. acerina* to get the final concentration of 20 µM and incubated at 150 r min⁻¹ and 20 °C. The medium (0.5 mL) was collected on the 3rd, 6th, and 11th days for chemical analysis. *n* = 5 biologically independent samples. The data are expressed as mean ± SD. n.s. not significant. **g–j** Time course analysis of PNL inoculated with *M. acerina*. Representative symptoms of infected PNL at 3, 5, and 14 DPI (**g**), absolute contents of Rb3 (left) and Fd (right) in Zone I (**h**), and their contents in Zone II (**i**) and Zone III (**j**) during a time course post the inoculation with *M. acerina*. **h–j** *n* = 4 and 5 biologically independent samples for mock and *M. acerina*-inoculated groups, respectively. The data are expressed as mean ± SEM. Statistical significance was calculated using Student's *t*-test. ***p* < 0.001; ****p* < 0.0001. Source data are provided as a Source Data file.

trypsin and subsequently analyzed by Nano LC-LTQ Orbitrap MS to construct a peptide library of accessible amino acid sequences. RNA sequencing of samples from the leaves, stems, and roots of *P. notoginseng* yielded 328 transcripts and 125 unigenes annotated as glycosidases. Given that ginsenoside hydrolysis occurs predominantly in PNLs, 33 transcripts and 19 unigenes that were highly expressed in the leaves were selected as a candidate gene library (Supplementary Fig. 3b). Then, mutual sequence alignment analysis between the peptide and transcript libraries was performed, resulting in the identification of three candidate genes (*PnGH1-3*) with a SEQUEST HT score >10 (Supplementary Fig. 3a and Supplementary Table 1), among which the protein encoded by *PnGH1* had the highest match score of 509.45.

Subsequent heterologous expression in *Escherichia coli* and purification via Ni²⁺ affinity chromatography allowed us to obtain the recombinant proteins PnGH1-3 (Fig. 3a, Supplementary Fig. 4). However, when these three recombinant proteins were individually incubated with Rb3 (**4**), only PnGH1 exhibited glucosidase activity resulting in the generation of **4a** (Fig. 3a). The molecule represented by this peak was unambiguously identified as Fd at the structural level through the analysis of its HRESI MS, 1D- and 2D-NMR spectra (Supplementary Fig. 5–9). The in vivo biochemical function of *PnGH1* was further investigated in *Nicotiana benthamiana* leaves via *Agrobacterium*-mediated transient expression (Supplementary Fig. 10), which also led to the generation of Fd after Rb3 was injected. PnGH1 remained active in a broad pH range from 3.0 to 9.0, and a temperature range from 10 to 80 °C (Supplementary Fig. 11). The kinetic analysis of PnGH1 under the identified optimal conditions (pH 5.0, 60 °C) indicated a $K_M = 677.0 \pm 38.5\,\mu M$, $k_{cat} = 3.9 \pm 0.05\,s^{-1}$, and $k_{cat}/K_M = 5.8 \times 10^{-3}\,s^{-1}\,\mu M^{-1}$ for Rb3 (Fig. 3a), demonstrating the high substrate affinity and considerable catalysis efficiency of PnGH1. Remarkably, PnGH1 could accept structurally diverse PPD-type

ginsenosides originating from *Panax* species, including Rb1 (**1**), Rc (**2**), Rb2 (**3**), Rb3 (**4**), Ra2 (**6**), Ra3 (**7**), Ra1 (**8**), Rd (**9**), and Rg3 (**10**), as substrates to produce the corresponding deglycosylated products (**1a–4a** and **6a–10a**) via the regioselective hydrolysis of the common β (1 → 2)-glycosidic linkage at C-3 (Supplementary Figs. 12, 13). In contrast, PnGH1 could not accept the PPT-type ginsenoside Rf or the flavonoid glycoside baimaside as a substrate (Supplementary Fig. 14).

The full-length cDNA of *PnGH1* contained a 1641-bp ORF encoding a 546-amino-acid protein, which was annotated as a glucosidase belonging to the GH1 family (clan GH-A) based on CAZyme analysis (http://www.cazy.org/) (Supplementary Fig. 15a). Phylogenetic analysis of PnGH1 and currently reported β-glucosidases involved in plant defence found that PnGH1 located at a distinct evolutionary branch compared with others (Supplementary Fig. 15b)[20,21]. Signal peptide prediction using iPSORT[22] revealed the presence of an *N*-terminal chloroplast transit peptide in PnGH1. Transient expression of a PnGH1-GFP fusion product in *Arabidopsis* protoplasts showed that PnGH1 is a chloroplast-localized glucosidase (Fig. 3b), which was consistent with the finding that ginsenoside hydrolysis occurs mainly in PNLs.

The structural basis of the regioselective hydrolysis ability of PnGH1 was further explored based on homology modeling (Supplementary Figs. 16, 17). The core structure of PnGH1 contained eight parallel β-strands surrounded by eight helices forming the $(\beta/\alpha)_8$ barrel fold as a substrate-binding pocket (Fig. 3c). In addition, several salt bridges between Asp/Glu residues and Lys/Arg residues were observed, which could make great contributions to the temperature tolerance of PnGH1 by stabilizing the protein structure. Further in silico docking and simulation (Supplementary Fig. 18) of Rg3 with PnGH1 revealed that the β-(1,2)-linked glycosidic bond at C-3 was positioned deep within the catalytic pocket, while the other side of the

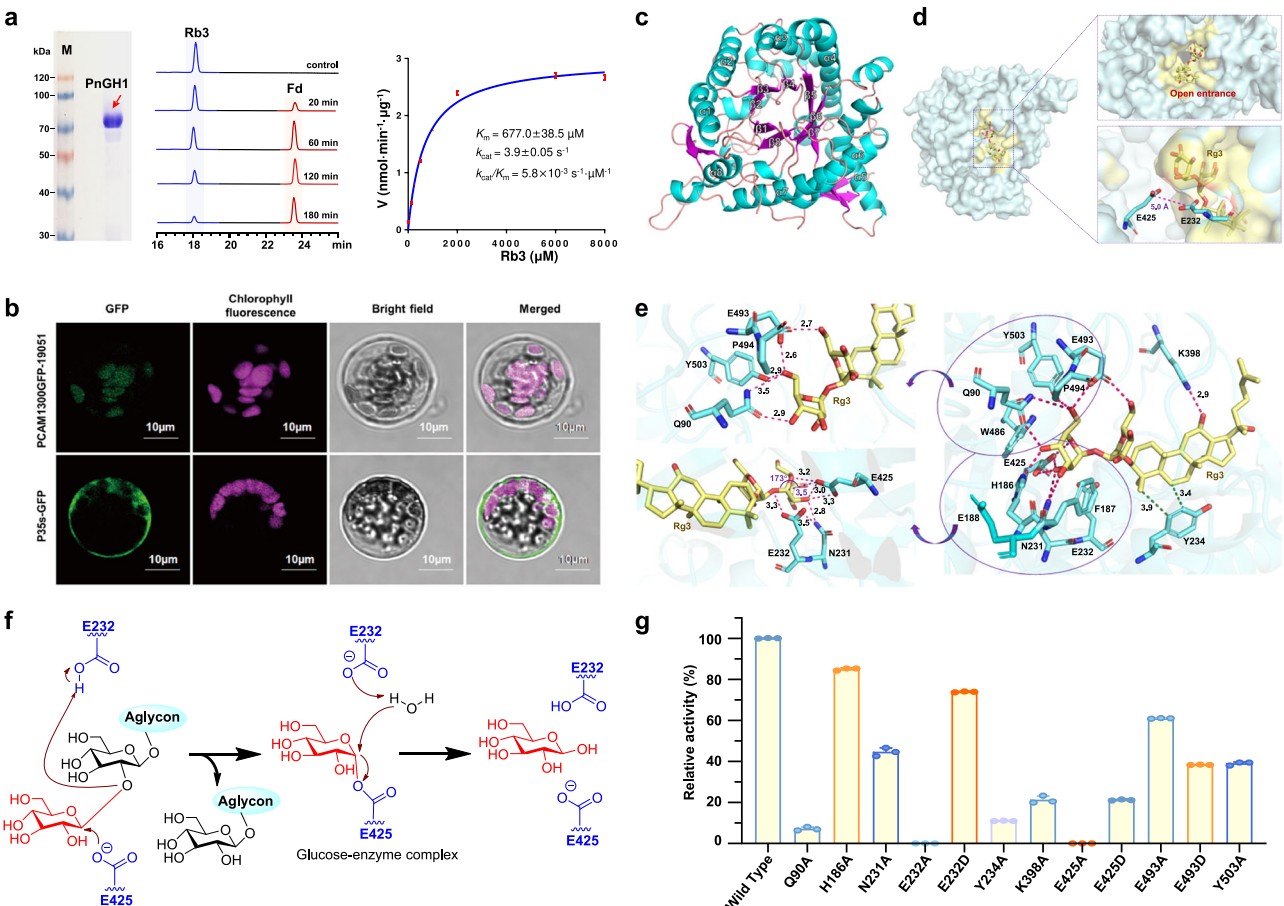

**Fig. 3 | Functional, biochemical characterization, and structural investigation of PnGH1. a** SDS–PAGE analysis, HPLC chromatogram, and kinetic study of the PnGH1-conducted hydrolysis reaction of Rb3 to form Fd. Kinetic assays were performed in independent duplicates. **b** Subcellular localization investigation of PnGH1. The experiment was repeated independently three times with similar results. **c** The overall structure of PnGH1 with the labeled $(\beta/\alpha)_8$ barrel. **d** Surface view of the binary complex structure of PnGH1 docking with Rg3 showing the widely open entrance of the substrate binding pocket. The distance between E425 and E232 (5.0 Å) was shown as a purple dash. **e** Docking and simulation results of key residues in ginsenosides substrates binding pocket of PnGH1. Rg3 and key residues were shown as yellow and cyan sticks, respectively. Hydrogen bonds and hydrophobic interactions were shown as purple and green dashes, respectively. **f** The retaining reaction mechanism of the hydrolysis of PPD-type ginsenosides catalyzed by PnGH1 through the double displacement mechanism. **g** The relative activity of different mutants of PnGH1. Data are presented as the mean ± SD of triplicate independent experiments. Source data are provided as a Source Data file.

substrate-binding pocket exhibited a wide, open entrance (Fig. 3d), making it possible for the protein to accommodate various PPD-type ginsenosides with different substituents at C-20. PnGH1 retained the glutamic acid-containing catalytic motifs TFN(E/D)P and (V/Y)ITENG, which are highly conserved in other known GH1 β-glucosidases (Supplementary Fig. 19)[20]. Comprehensive analysis of the amino acid residues lining the catalytic pocket (Fig. 3e) revealed that the critical residues E232 in the TFNE/DP motif and E425 in the (V/Y)ITENG motif were indeed situated on opposite sides of the $\beta$-(1,2)-linked glycosidic bond, indicating a distance of 3.5 Å between the OE2 atom of E425 ($O_{nucleophile}$) and the anomeric carbon ($C_{electrophile}$) and 3.3 Å between the OE2 atom of E232 and the oxygen atom of the leaving sugar group ($O_{leaving}$). The measured $O_{nucleophile}$–$C_{electrophile}$–$O_{leaving}$ angle was 173° (Fig. 3e). In addition, the distance between E232 and E425 was 5.0 Å (Fig. 3d), suggesting that PnGH1 is a retaining glucosidase rather than an inverting enzyme, in which the distance between these two catalytic residues would be ~10.0 Å[23]. The above results suggest that PnGH1 employs the well-studied acid-catalytic retaining mechanism[23] requiring the involvement of two critical carboxylate residues, E232 and E425 (Fig. 3f). Accordingly, the mutagenesis of E232 and E425 to alanine deactivated the hydrolysis activity of PnGH1. In contrast, when E232 was replaced with aspartic acid, the enzyme activity could be effectively recovered (Fig. 3g). This observation further indicated the

crucial role of carboxylate residues in the hydrolysis process. Docking results showed that several other residues, in addition to E232 and E425, were located in the catalytic pocket of PnGH1 and interacted with the substrate (Fig. 3e, Supplementary Table 2) through hydrogen-bond interactions (including Q90, H186, N231, K398, E425, E493, and Y503) and hydrophobic interactions (Y234). Alanine screening mutagenesis of these residues also led to an apparent decrease in enzymatic activity (Fig. 3g), which further supported their participation in the hydrolysis process.

## PnGH1 catalyzes the hydrolysis upon chloroplast disruption caused by induced exoenzymes of *M. acerina*

To reveal how chloroplast-localized PnGH1 triggers hydrolysis of ginsenosides that are possibly localized in the cytoplasm, we first investigated whether ginsenoside hydrolysis was activated in PNLs upon mechanical tissue damage. Homogenization with a bead mill was found to achieve adequate cell disruption, mimicking the tissue damage caused by attackers[24]. PnGH1 substrates (**1–4**) in fresh healthy PNLs were entirely hydrolyzed to their corresponding products (**1a–4a**), as measured by HPLC-UV (Fig. 4a), indicating that ginsenoside hydrolysis occurs in healthy PNLs upon tissue damage.

To determine whether fungal infection caused chloroplast disruption in PNLs, the ultrastructure of infected PNLs was observed by

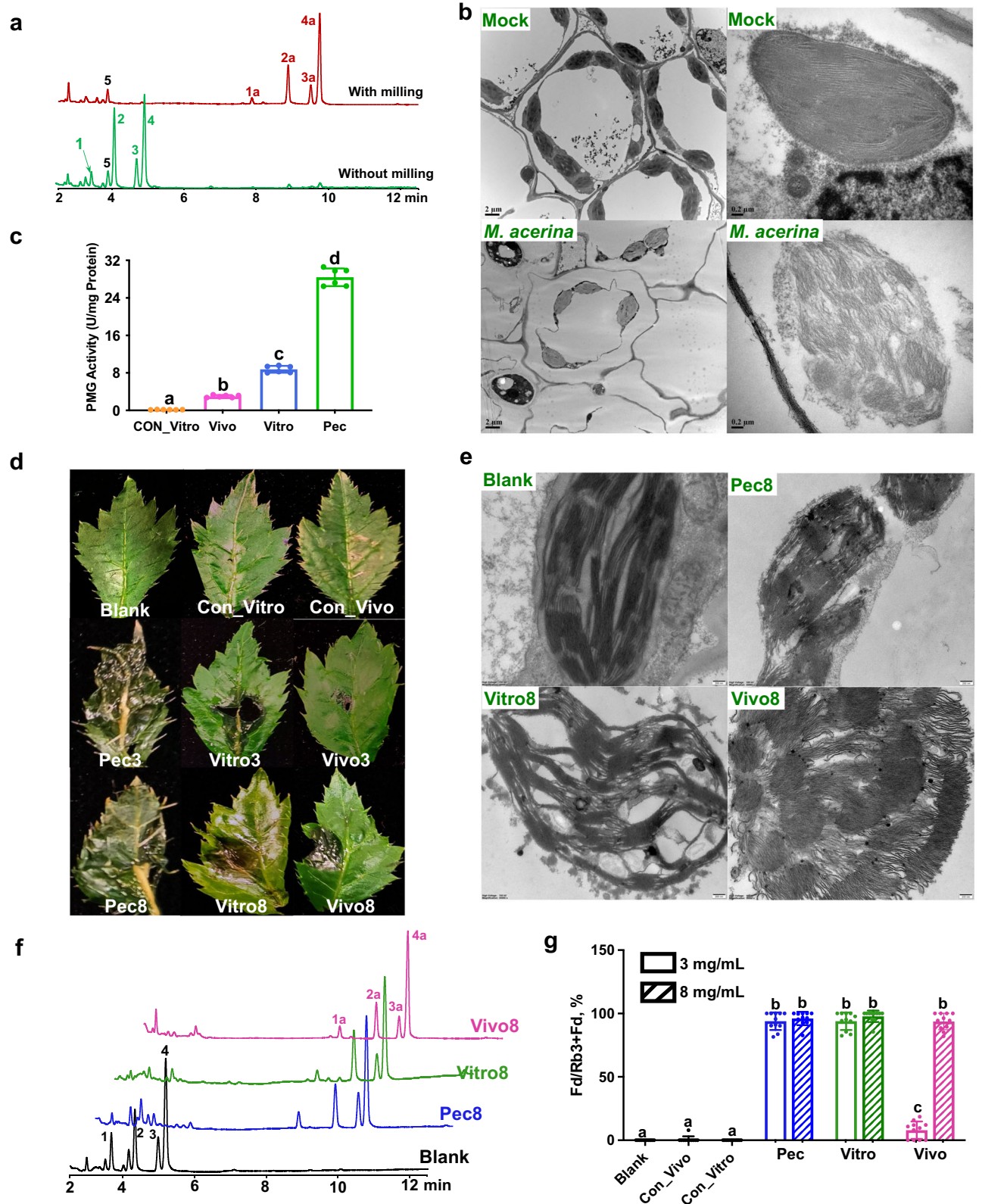

TEM. In mock-treated PNLs, chloroplasts with an elliptical or lens-like shape were arranged along the plasma membrane, with distinct grana stacking and an intact chloroplast envelope (Fig. 4b). Infection with *M. acerina* caused cell integrity loss, severe plasmolysis, and organelle disintegration of various degrees in the PNLs. Severe chloroplast degradation was observed, including disruption of the chloroplast membrane and degradation of grana and stroma thylakoids (Fig. 4b),

indicating that fungal infection causes chloroplast membrane disruption, following by the activation of ginsenoside hydrolysis.

Extracellular enzymes, especially pectolytic enzymes, play a crucial role in the fungal colonization of plant hosts[25]. To further investigate how *M. acerina* infection causes cell chloroplast membrane disruption in PNLs, crude exoenzymes of *M. acerina* whose expression was induced by exposure to PNL tissue (both in vivo and in vitro) or

**Fig. 4 | PnGH1 catalyzes ginsenoside hydrolysis upon chloroplast disruption caused by induced-exoenzymes of *M. acerina*. a** HPLC-UV analysis of healthy PNL with (above) or without (below) homogenization by bead milling. **b** Representative transmission electron microscopy (TEM) micrographs of PNL inoculated by PDA disc with and without *M. acerina* at 5 DPI (left, scale bars = 2 μm; right, zoomed chloroplast, scale bars = 0.2 μm). **c** Pectin methylgalacturonase (PMG) activities, *n* = 5 biologically independent samples. **d** Photographs of representative symptoms in PNL inoculated with controls and crude exoenzyme induced by PNL in vivo and in vitro, and pectin (Pec) at the concentration of 3 and 8 mg mL⁻¹ at 5 DPI. Leaves treated with sterile water were used as blank control, the crude exoenzyme extracted from the steamed PNL and *M. acerina*-inoculated medium without pectin or PNL tissue were used as in vivo control (Con_vivo, 8 mg mL⁻¹ and in vitro control (Con_vitro, 8 mg mL⁻¹), respectively. **e–g** representative TEM images (**e**), HPLC-UV profile (**f**), and hydrolysis turnovers (**g**) of PNL inoculated with the crude exoenzymes induced by PNL or pectin. **1**, Rb1; **2**, Rc; **3**, Rb2; **4**, Rb3; **5**, Fc, **1a**, XVII; **2a**, Fe; **3a**, Rd2; **4a**, Fd. **g** *n* = 11 biologically independent samples for each group. **c**, **g** The data are expressed as mean ± SD. Statistical significance was calculated using a one-way ANOVA, followed by Tukey's post hoc test. Data with different letters above the bars are significantly different (*p* < 0.05). The experiments in **b–e** were repeated independently at least three times with similar results. Source data are provided as a Source Data file.

pectin (induced exoenzymes) were obtained, and their effects on plant leaves were investigated by the inoculation of PNLs with the exoenzymes (Supplementary Fig. 20a). The crude exoenzymes induced by PNLs and pectin promoted significantly higher polymethyl galacturonase (PMG, pectin hydrolase) activity than the broth culture of *M. acerina*, and the highest activity of PMG was found in medium containing pectin (Fig. 4c). The blank (distilled water) and control samples, including steamed PNLs without fungal inoculation (Con_vivo, 8 mg mL⁻¹) and *M. acerina*-inoculated medium without pectin or PNL tissue (Con_vitro, 8 mg mL⁻¹), failed to produce any lesions on PNLs. However, the inoculation with crude exoenzymes at 3 and 8 mg mL⁻¹ using absorbent cotton led to leaf wilting (Supplementary Fig. 20b) and induced severe necrotic lesions in the leaves at 4 DPI (Fig. 4d), especially in pectin-induced samples, indicating the possible involvement of induced exoenzymes in necrotic lesion development.

The ultrastructure of exoenzyme-inoculated leaves observed by TEM indicated chloroplast degradation and membrane disruption (Fig. 4e). The chemical analysis of decayed tissue also indicated that PnGH1 substrates were completely hydrolyzed after inoculation with induced exoenzymes, except in the in vivo group treated with a low concentration (3 mg mL⁻¹, Fig. 4f, Supplementary Fig. 20b). A similar result was observed for the hydrolysis turnover (Fig. 4g). Collectively, the symptoms, chloroplast ultrastructure and chemical changes observed in the leaves treated with induced exoenzymes were similar to those caused by infection with *M. acerina*, suggesting that PnGH1 catalyzes ginsenoside hydrolysis upon chloroplast membrane disruption caused by induced exoenzymes of *M. acerina*.

## Hydrolysis products participate in plant chemical defence by inhibiting pathogen growth

To evaluate the possible biological significance of PnGH1-mediated ginsenoside hydrolysis for plant defence, the toxic effects of intact ginsenosides and their hydrolysis products against fungal growth were compared in vitro and in vivo. Refined total saponin extracts of PNLs before (TS-b) and after (TS-a) hydrolysis, mainly containing intact ginsenosides and hydrolysis products, respectively, were prepared and purified by transformation and using microporous resins[26] (Supplementary Fig. 21a). TS-b showed weak inhibitory activity against *M. acerina* growth, whereas growth inhibition by TS-a was dose-dependent and reached 80.8% at 4 mg mL⁻¹. TS-a exhibited significantly stronger inhibitory effects against *M. acerina* growth at the applied concentrations (0.13–4 mg mL⁻¹) than TS-b (Fig. 5a, b). A similar tendency was observed in *A. panax* (Fig. 5c, Supplementary Fig. 21b) and *F. oxysporum* (Supplementary Fig. 21c, d), indicating that refined extracts containing hydrolysis products are more toxic to a broad spectrum of fungi at the tested concentration ranges than intact ginsenosides. The inhibition of *M. acerina* growth after exposure to a series of concentrations of individual intact ginsenosides (i.e., Rb3, Rc, and Rb1) and their hydrolysis products (i.e., Fd, Fe, and XVII, respectively) was also examined. As expected, Fd, Fe, and XVII exhibited more potent inhibitory effects against *M. acerina* than their intact ginsenosides Rb3, Rc, and Rb1, respectively (Fig. 5d, Supplementary Fig. 21e–g). Interestingly, Fd, the most abundant component

in decayed PNLs, exhibited the most potent antifungal activity against *M. acerina* (Fig. 5d).

Next, the susceptibility of PNLs to *M. acerina* after the exogenous spraying of a solution of intact ginsenosides and their hydrolysis products was evaluated in pot experiments. TS-a significantly alleviated the severity of round spot disease caused by *M. acerina* in the PNLs compared with that in the control and TS-b groups. The lesion area exhibited a TS-a dose-dependent decrease (Fig. 5e). The control efficacy of TS-a at a dose of 4 mg mL⁻¹ was more potent than that of carbendazim (200 μM), used as a positive control (Fig. 5e). Additionally, compared with Rb3 spraying, Fd spraying significantly decreased the necrotic lesion area on the PNLs, except at a low dose of 0.5 mM (Fig. 5f). Collectively, these results show that compared with intact ginsenosides, their hydrolysis products exert more potent antifungal activity both in vitro and in vivo, suggesting that the plants may respond to pathogen attack by increasing the production of toxic compounds via PnGH1-mediated ginsenoside hydrolysis.

## A two-component defence system exists in congeneric *Panax* species

To determine whether a similar two-component system exists in other congeneric *Panax* species, *P. quinquefolium* leaves (PQLs) with black spot disease caused by *A. panax* in the open-field, and PQLs and *P. ginseng* leaves (PGLs) inoculated with *A. panax* in the greenhouse were collected and analyzed by HPLC-UV. Rb3 and Rd were the most abundant components involved in ginsenoside hydrolysis in PQLs and PGLs, respectively (Fig. 6). Rb3 was mainly hydrolyzed to Fd in infected tissue of both PQLs with black spot disease (Fig. 6a) and *A. panax*-inoculated PQLs (Fig. 6b), and Rd was hydrolyzed to F2 in the infected tissue of *A. panax*-inoculated PGLs (Fig. 6c), indicating that ginsenoside hydrolysis with similar pattern occurs in pathogenic fungus-infected leaves of *P. quinquefolium* and *P. ginseng*.

Subsequent searches for homologous genes to *PnGH1* in the *P. ginseng* genome database and the *P. quinquefolium* transcriptome database led to the discovery of two homologous genes in *P. ginseng*, *PgGH1*, and *PgGH2*, showing more than 93% sequence identity with *PnGH1* and one homologous gene in *P. quinquefolium*, *PqGH1*, with 90% sequence identity to *PnGH1*. iPSORT prediction showed that an *N*-terminal chloroplast transit peptide was also present in PgGH1, PgGH2, and PqGH1, suggesting that all these glucosidases had a similar chloroplast subcellular localization. Heterologous expression of these enzymes in *E. coli* (Fig. 6d) and enzymatic reaction experiments indicated that PgGH1, PgGH2, and PqGH1 also showed *β*-glucosidase activity that catalyzed the efficient transformation of Rb3 to Fd (Fig. 6e). These results suggest that *β*-glucosidase-mediated ginsenoside hydrolysis occurs in multiple *Panax* species and significantly contributes to the chemical defence processes of these plants.

## Discussion

Ginsenosides are ubiquitous secondary metabolites found in *Panax* plants in species- and organ-specific manner. *P. notoginseng* plants use considerable resources to synthesize these ginsenosides, implying that the compounds have essential plant functions or benefits.

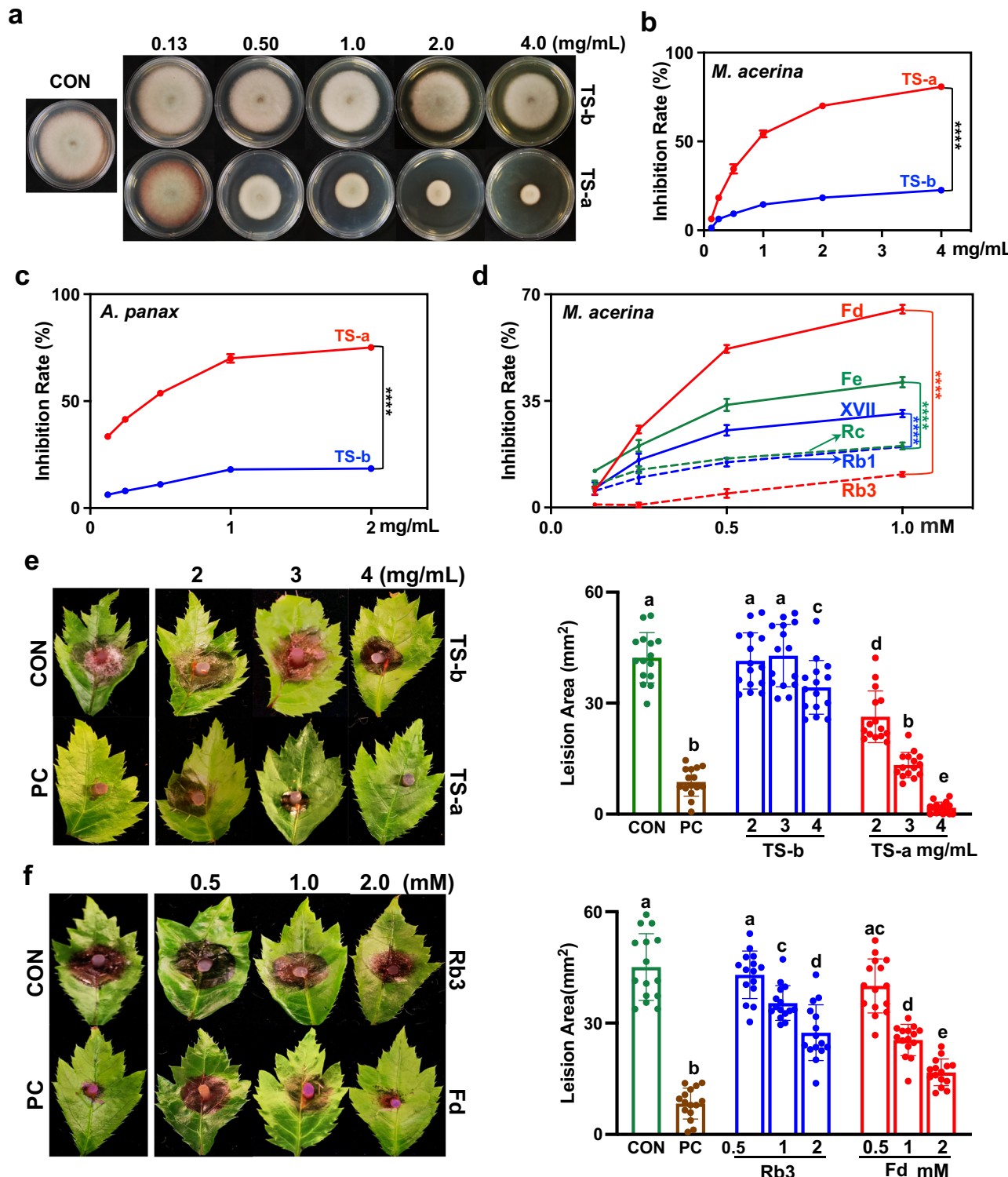

Ginsenosides have attracted increasing attention in the past two decades due to their significant medicinal value, but few studies have focused on their roles in plant defence. In this study, we found that *P. notoginseng* has evolved a two-component chemical defence system composed of PnGH1, a glucosidase localized in chloroplasts, and PPD-type ginsenosides with $\beta$-1,2-glucosidic linkages at C-3. This two-component defence system might be crucial in chemical defence against fungal infection in *P. notoginseng*.

The round spot is a primary foliar destructive disease occurring in cultured *P. notoginseng* caused by the necrotrophic fungus *M.*

*acerina*[27,28], leading to 50–90% yield loss[27]. In our study, the first indication that PPD-type ginsenosides have defensive properties against fungal infection was that ginsenoside hydrolysis was localized to the decayed tissue of PNLs with round spot disease. These findings were also confirmed in *M. acerina*-inoculated PNLs in the greenhouse. Nevertheless, a small quantity of hydrolysis products was detected in the leaves of non-infected plants and healthy tissue in the same and neighboring leaves, indicating that other unidentified factors or stresses might also cause mild ginsenoside hydrolysis in healthy PNLs under highly complex open-field conditions. However, upon fungal

**Fig. 5 | Hydrolysis products participate in plant chemical defence by inhibiting pathogen growth in vitro and in vivo. a, b** Hydrolysis products suppressed the mycelial growth of *M. acerina* in a dose-dependent manner. Inhibitory effects of refined total saponins of PNL before (TS-b) and after (TS-a) the hydrolysis on colony diameter at 5 DPI (**a**) and corresponding growth inhibition rate (**b**, *n* = 6). **c** Growth inhibition rate of refined total saponins against *A. panax*. *n* = 5 biologically independent samples. **d** Growth inhibition ratios of individual ginsenosides and hydrolysis products against *M. acerina*. A mycelial disc (5 mm) was punched from the actively growing edge of the fungal PDA plates and inoculated in the center of PDA plates containing different concentrations of tested total saponins or ginsenosides. Each colony diameter was measured at 5 DPI by a cross-bracketing method. *n* = 4 biologically independent samples. The data are expressed as mean ± SD. Statistical significance was calculated using Student's *t*-test.

****$p < 0.0001$. **e** and **f** in vivo antifungal activity (left) and comparison of lesion area (right) of total saponins (**e**) and Rb3/Fd (**f**) against *M. acerina*. The solutions containing different concentrations of total saponins or Rb3/Fd were sprayed on the surface of PNL 3 times per day for 3 consecutive days. On the 4th day, the mycelial plugs of *M. acerina* were inoculated on the previously processed PNL. After 5 DPI, the infected leaves were photographed, and the lesion area was calculated. Carbendazim (200 μM) and blank solution were used as the positive control (PC) and negative control (CON), respectively. *n* = 15 biologically independent samples. The data are expressed as mean ± SD. Statistical significance was calculated using a one-way ANOVA, followed by Tukey's post hoc test. Data with different letters above the bars are significantly different ($p < 0.05$). Source data are provided as a Source Data file.

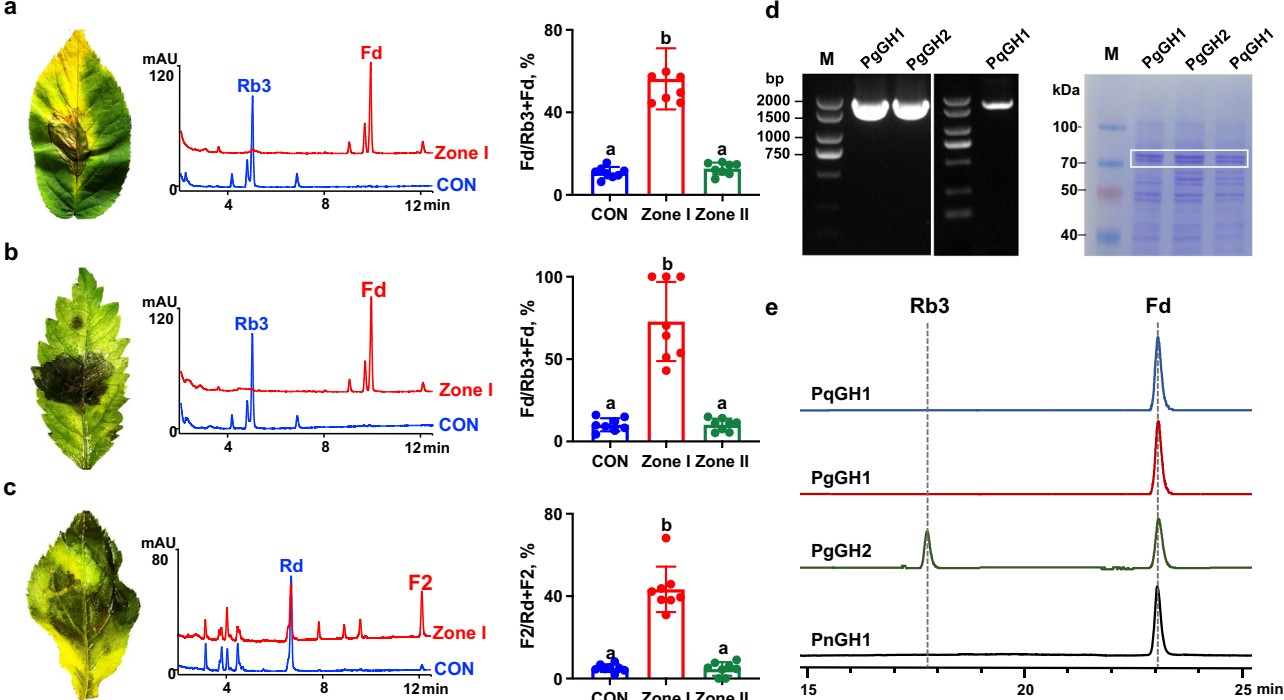

**Fig. 6 | Ginsenoside hydrolysis catalyzes by $\beta$-glucosidase in *P. quinquefolium* leaves (PQL) and *P. ginseng leaves* (PGL). a** PQL with black spot disease caused by *A. panax* collected in the open field. **b** PQL inoculated with *A. panax* at 5 DPI in the greenhouse. **c** PGL inoculated with *A. panax* at 5 DPI. Representative symptoms of infected leaves (left), HPLC-UV chromatograms of methanolic extracts (center) of healthy leaves (CON) and decayed tissue (Zone I), and hydrolysis turnovers of different zones in leaves (left), calculated as the ratio of Fd/Fd + Rb3 (for PGL) or F2/Rd + F2 (for PQL). **a–c** *n* = 8 biologically independent samples. The data are

expressed as mean ± SD. Statistical significance was calculated using a one-way ANOVA, followed Tukey's post hoc test. Data with different letters above the bars are significantly different ($p < 0.05$). **d** Gene cloning and heterogenous expression of *PgGH1*, *PgGH2* from *P. ginseng*, and *PqGH1* from *P. quinquefolium*. The targeted proteins were labeled in the frame. The experiment was repeated independently three times with similar results. **e** HPLC chromatograms of hydrolysis reactions of Rb3 to Fd catalyzed by PnGH1, PgGH1, PgGH2 and PqGH1, respectively. Source data are provided as a Source Data file.

infection, intact ginsenosides are largely hydrolyzed in the decayed tissue of the PNLs. We further revealed that ginsenoside hydrolysis is a stress type-specific response associated with infection by pathogenic fungi, but absent in non-pathogenic fungal infection and designed abiotic stress. Dynamic chemical changes during fungal infection indicated that the molar amounts of the hydrolysis products in decayed PNLs increased dramatically, with increases ranging from 2.2-fold to 7.8-fold, suggesting that ginsenoside biosynthesis is enhanced to produce greater amounts of the hydrolysis products upon fungal infection. Fd, with the strongest antifungal activity, showed the most significant enrichment during the infection process. Gene expression profile analysis revealed that the expression of the key genes involved in the biosynthesis of ginsenosides and the $\beta$-glucosidase gene *PnGH1* upregulated significantly after being infected by the necrotrophic fungus *M. acerina* (Supplementary Fig. 22). Jasmonates (JAs) are a

group of lipid-derived phytohormones that play an essential role in a variety of stress defence responses and plant growth[29]. Treatment with methyl jasmonate (MeJA) effectively enhances Rb1, Rc, Rb2, and Rb3 biosynthesis by activating gene transcription related to ginsenoside biosynthesis in PNLs[30]. The content of Rg1, Re, Rb1, and Rd is significantly increased in *P. notoginseng* cell suspension cultures upon 2-hydroxyethyl jasmonate (HEJ) or MeJA treatment[31]. Thus, JAs are important signaling molecules that elicit ginsenoside biosynthesis in *Panax*. The inoculation with pathogenic fungi results in upregulating JA biosynthesis-related genes and JA signaling pathway genes in PNLs[32]. In our study, the enhanced ginsenoside biosynthesis during fungal infection might be attributed to the activation of the JA signaling cascade.

Two-component defence systems involving several classes of specialized metabolites, including cyanogenic glycosides, glucosinolates,

iridoids, benzoxazinoids, and triterpene saponins, have long been documented in other genera[20,21]. Triterpene saponins exert antifungal activity by inhibiting biofilm formation and disrupting the membrane integrity by forming saponin-sterol complex, acting as plant defence compounds[20,33]. Oat (*Avena sativa*) leaves accumulate non-active bis-desmosidic saponins that can be bio-activated by a specific β-glucosidase, avenacosidases, to exert their biocidal effects[34]. Recently, Lacchini et al. reported that *Medicago truncatula*, a model legume, has evolved a two-component chemical defence system composed of triterpene saponins and a nucleolar-localized β-glucosidase, acting as a saponin bomb[21]. Thus, the saponin bomb model might be a widespread two-component system in many plant species. However, such a system has never previously been reported for ginsenosides in *Panax* species. Less polar ginsenosides lead to increased antifungal activity, indicating the importance of sugar moieties in toxicity toward fungi[35]. Our data indicated that PnGH1-mediated hydrolysis products, either in the form of refined extracts or pure compounds, exhibited more potent broad-spectrum antifungal activity against phytopathogens both in vitro and in vivo than intact ginsenosides. Hydrolysis products with lower polarity might more easily cross the membrane and be enriched in the fungus to exert more potent fungicidal activity. Previous studies have indicated that triterpenoid saponins are localized in the cytoplasm[8,36] or vacuoles[21,37] of plant cells, but little information is available regarding ginsenoside localization in PNLs. The identified hydrolysis products with strong antimicrobial activity were consistent with the definition of phytoalexins, which are inducible specialized metabolites that are constitutively produced in response to biotic stresses.

As bioactivating enzymes, β-glucosidases are often involved in such defence systems. In this study, PnGH1, a β-glucosidase with high enzyme stability and hydrolytic regiospecificity, was discovered in *P. notoginseng* and characterized. The $k_{cat}/K_M$ value of PnGH1 ($5.8 \times 10^{-3}\,s^{-1}\,\mu M^{-1}$ for Rb3) significantly surpasses that of MeLinamarase ($0.0246 \times 10^{-3}\,s^{-1}\,\mu M^{-1}$ for linamarin)[38], a cyanogenic β-glucosidase involved in plant defence through cyanogenesis mechanism against small herbivores, and are comparable in terms of magnitude to that of MtG1 ($26.3 \times 10^{-3}\,s^{-1}\,\mu M^{-1}$ for 3-Glc-28-Glc-medicagenic acid)[21], a triterpene saponin β-glucosidase involved in a two-component chemical defence system in *M. truncatula*. Modeling and mutagenesis studies suggested that PnGH1, belonging to the GH1 family, is a retaining β-glucosidase that hydrolyzes glycosidic bonds through a double displacement mechanism using two conserved carboxylic acid residues, E232 and E425, which serve as the catalytic acid/base and nucleophile, respectively. Thus far, PnGH1 appears to be a unique β-glucosidase that can hydrolyze an outer glucose from the β $(1 \rightarrow 2)$-glycosidic linkage at C-3 of PPD-type ginsenosides. *PnGH1* is expressed abundantly in PNLs. The *N*-terminus of the PnGH1 protein contains a chloroplast transit peptide, which is consistent with the identified subcellular localization of PnGH1 in the chloroplast. Upon cellular disruption caused by an event such as a pathogen attack, toxic defence compounds are bioactivated by removing a glycosidic moiety catalyzed by β-glucosidase, which is spatially separated from protoxin substrates under physiological conditions.

PnGH1 was shown to subcellularly localize to the chloroplast, which insulated it from its substrates in intact cells. This is a critical characteristic of a two-component defence system, which relies on the physical separation of β-glucosidase from protoxin substances. PnGH1 triggers hydrolysis upon cell disruption in fresh healthy PNLs, as demonstrated by a bead milling experiment. A similar result was found in an aqueous extract of dried PNLs following ultrasonication[10,18], an efficient method for disrupting cell walls and membranes[39]. Furthermore, massive cell disruption can be elicited by fungal infection. Plant necrotrophic pathogens secrete a wide range of cell wall-degrading enzymes or exoenzymes to deconstruct the structural components of plant cells, facilitating penetration and colonization of host tissues[25]. Consistent with this, our data indicate that infection with *M. acerina* causes organelle disintegration and chloroplast architecture degradation in PNLs, as observed by TEM, resulting in the release of PnGH1 from the chloroplast. Mechanistically, the exoenzymes whose expression is induced by this fungus by PNL tissue and pectin contribute greatly to cell disruption, as evidenced by the observation that the symptoms, chloroplast ultrastructure, and chemical changes in the leaves treated with induced exoenzymes were similar to those caused by infection with *M. acerina*, although the exoenzymes from a broth culture of *M. acerina* failed to produce any lesions on the PNLs.

In conclusion, this work clearly demonstrates a two-component defence system comprising the β-glucosidase PnGH1 and PPD-type ginsenosides in *P. notoginseng* (Fig. 7). Similar ginsenoside hydrolysis was observed in pathogenic fungus-infected leaves of *P. quinquefolium*, and *P. ginseng*, and their responsible homologous β-glucosidases were also cloned and functionally identified. Our findings reveal a two-

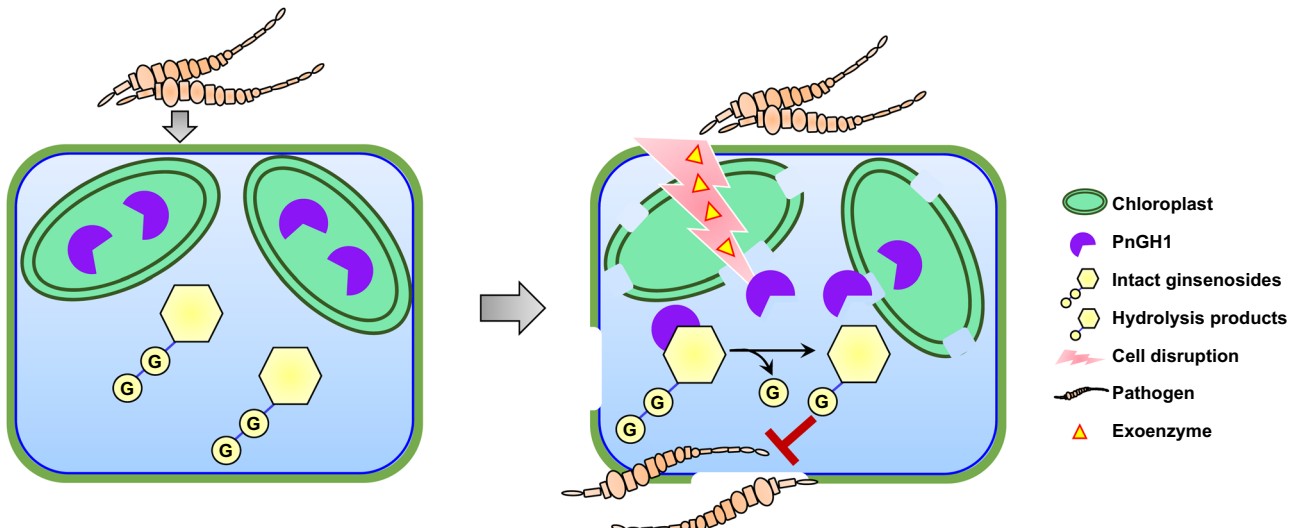

**Fig. 7 | Schematic illustration representing PnGH1-mediated ginsenoside hydrolysis in *P. notoginseng* leaves and its role in chemical defence against pathogenic phytopathogens.** *P. notoginseng* has evolved a two-component defence system comprising the β-glucosidase PnGH1 and PPD-type ginsenosides. Chloroplast-localized PnGH1 is released and mixed with its substrates upon cell disruption, potentially mediated by the induced exoenzymes of pathogenic fungi. Subsequently, PnGH1-mediated hydrolysis produces a series of less-polar ginsenosides with more potent antifungal activity.

component chemical defence system in *Panax* species and offer insights for developing botanical pesticides for disease management in *Panax* species.

## Methods

### Chemical and materials
Ginsenoside standards, i.e. Rb1 (**1**), Rc (**2**), Rb2 (**3**), Rb3 (**4**), XVII (**1a**), Fe (**2a**), Rd2 (**3a**), Fd (**4a**) and Fc (**5**), were purchased from commercial companies, and their purities were above 98% as measured by HPLC-UV. The pectin and pectinase were obtained from Millipore Sigma (Sigma-Aldrich, St. Louis, MO, USA). *P. notoginseng* leaves (PNL), *P. ginseng* leaves (PGL), and *P. quinquefolium* leaves (PQL) were collected during 2019–2022 from Yunnan and Jilin provinces of China. They were authenticated by Prof. Xia-Hong He from Yunnan Agricultural University, and their voucher species were deposited in the Institute of Chinese Medical Sciences, University of Macau.

### Plant growth
For the field study, *P. notoginseng* seeds were germinated in a nursery garden and ecologically cultivated under forest conditions in Pu'er City, Yunnan Province. Field experiments were performed in the rainy season at the *P. notoginseng* cultural fields to evaluate chemical changes in the leaves upon various abiotic and biotic stresses. Five field experiments were conducted in Lancang Country (Pu'er City, Yunnan Province), a pesticide- and chemical fertilizer-free environment. The location information of five fields was listed as follows: (i) Laomianxinzhai (99°51′50″E, 22°38′41″N, 1630 m); (ii) Xiaoguangzha (99°50′41″E, 22°40′36″N, 1574 m); (iii) Datangzi (99°47′26″E, 22°42′16″N, 1467 m); (iv) Miaoxiangsanqi (99°49′56″E, 22°39′43″N, 1710 m); (v) Xueyuandi (99°78′16″E, 22°80′39″N, 1481 m). For lab inoculation experiments, one-year-old *P. notoginseng* was grown under a rain shelter covered with double sunshade nets at an experimental base of Yunnan Agricultural University, Kunming, China. In the growing season, *P. notoginseng* was moved to the greenhouse for adaptation for 2 weeks and disinfected with sterile water. For the anti-pathogen test in vivo and exoenzyme inoculation experiment, the seeds of *P. notoginseng* sprouted in a moisturizing environment and were transplanted into a culture bottle containing Murashige & Skoog Medium (MS medium, without agar and sucrose) and vermiculite.

### Sample preparation and chemical analysis
Different positions of PNL from the plants upon various abiotic and biotic stress were harvested for chemical analysis as described in Fig. 1a. The lesion area (Zone I), the healthy area in the same leaf (Zone II) and the healthy area in the neighboring leaf (Zone III) were collected by a 5-mm hole puncher, and immediately soaked in 500 μL methanol. Sample preparation was conducted by ultrasonication with slight modifications, as reported previously[10]. The leaf tissue was cut into pieces and ultrasonically extracted (135 W and 44 kHz) with 500 μL methanol at room temperature for 30 min, then centrifuged at $4700 \times g$ for 15 min. The supernatant was filtered through a 0.22 μm polyvinylidene difluoride syringe filter (Carrigtwohill, Cork, Ireland) and analyzed using Method 1 in Supplementary Table 3.

### Activity-guided protein purification
*P. notoginseng* leaves were extracted by water through ultrafiltration (60 min, 3 times) to remove the saponins. The filtered pellet (100 g) was then homogenized in 1.5 L protein lysis buffer (50 mM PBS buffer, pH 7.4, 1 mM EDTA, 3 mM $\beta$-mercaptoethanol, 1 mM phenylmethanesulfonyl fluoride) at 4 °C. The supernatant was obtained as the crude enzyme solution after centrifugation at $7500 \times g$, 4 °C for 30 min. Next, ammonium sulfate fractional precipitation was performed at 30%, 50%, 70%, and 90% saturation, respectively. The obtained protein concentrate was desalted using a PD-10 column (GE Healthcare, Sweden) in 50 mM Tris–HCl buffer (pH 7.4) for further chromatographic separation.

The whole purification procedure was performed using an FPLC system (AKTA purifier, GE Electric) at 0–4 °C. Hydrolysis activities of the eluted fractions were tested through in vitro enzymatic assays containing 50 mM Tris–HCl, pH 7.4, 500 μM Rb3 as substrate, and 10 μg of protein fractions in a final volume of 100 μL. The reactions were incubated at 50 °C for 4 h and then terminated by adding 200 μL of ice-cold MeOH. After centrifuging at $12,000 \times g$ for 30 min, the supernatants were analyzed using Method 2 in Supplementary Table 3.

Crude enzyme solutions were initially submitted to anion exchange chromatography on a HiTrap Q HP column (1.6 cm × 2.5 cm, 8.6 μm, 5 mL, GE Healthcare). In the first round, 50 mM Tris–HCl buffer at pH 7.4 was used as the equilibration buffer. Targeted enzymes were concentrated in the unbound fractions. In the second round, 50 mM Tris–HCl buffer at pH 9.0 was used. Proteins were eluted with a linear 0–1 M NaCl gradient at a flow rate of 5 mL min⁻¹. The obtained active fractions were then concentrated in 5 kDa microcentrifuge filters and subjected to further two-round gel filtration purification using a Superdex 75 Increase 10/300 GL (10 mm × 300 mm) column. In the first round, protein samples were eluted with 50 mM PBS buffer at a flow rate of 0.8 mL min⁻¹. The highest active fraction was concentrated in 5 kDa microcentrifuge filters and then transferred to the second round of gel filtration under the same separation condition.

### Tryptic peptide LC–MS/MS analysis and comparative transcriptome analysis of *P. notoginseng*
The protein band on the SDS–PAGE gel was destained and resuspended in 200 μL 8 M urea/100 mM ammonium bicarbonate, and reduced with 10 μL 200 mM DTT for 15 min at 65 °C. Then the mixture was alkylated with 15 μL 500 mM iodoacetamide and incubated with a gentle shake at room temperature in the dark for 30 min. The supernatant was removed by centrifugation at $13,000 \times g$, and the beads were washed thrice with 50 mM ammonium bicarbonate. The digest was conducted using 1 μg sequencing-grade trypsin (10 ng μL⁻¹ trypsin, 50 mM ammonium bicarbonate, pH 8.0) with 1 μL 100 mM CaCl₂. Tryptic peptide sequences were analyzed on a Nano-LC-LTQ Orbitrap Velos Pro MS (ThermoFisher Scientific) using Method 3 in Supplementary Table 3.

Comparative transcriptome analysis was performed using root, stem, and leaf samples randomly collected from healthy *P. notoginseng* plants. Total RNA was isolated using a Plant RNA extraction kit (OMEGA, USA). Transcriptome sequencing and analysis were completed using the BGISEQ-500 sequencing platform (BGI, Wuhan, China). The fragments per kb per million reads (FPKM) method was employed to quantify the transcript abundances.

For candidate genes search, the raw MS data of the tryptic peptide sequences were blasted against the transcriptome database of *P. notoginseng* using Proteome Discoverer Software V.1.4.

### Gene cloning and heterologous expression of *PnGH1*–*PnGH3* from *P. notoginseng*
The coding region of *PnGH1*–*PnGH3* was amplified from cDNA of *P. notoginseng* using the primer sets *PnGH1*–*PnGH3*-F/R (Supplementary Table 4). Insertion of the *PnGH* genes into the linearized vector pET-32a was performed with a ClonExpress II One Step Cloning Kit (Vazyme, China) using *Eco*R I and *Xho* I as restriction sites according to the manufacturer's protocol. The recombinant plasmids were then introduced into *E. coli* Transetta DE3 (TransGen, China) and confirmed by sequencing. Recombinant strains were grown at 37 °C, 200 rpm in LB medium with 50 mg L⁻¹ ampicillin and 40 mg L⁻¹ chloramphenicol until the $OD_{600}$ value reached 0.4–0.6. Protein expression was induced by adding 0.5 mM isopropyl $\beta$-D-thiogalactopyranoside (IPTG). Strains were grown for 16 h at 18 °C, 200 rpm, and harvested by centrifugation at $7000 \times g$ for 20 min. The obtained cell pellets were suspended in lysis buffer (20 mM phosphate buffer, 20 mM imidazole, 500 mM NaCl, 0.5 mM phenylmethylsulfonyl fluoride, 10% glycerol, pH 7.4).

After sonication (800 bar, 3 times), the sample was centrifuged at $9500 \times g$, 4 °C for 30 min, and the supernatant was filtered through a 0.45 μm membrane. A pre-equilibrated Histrap column (GE Healthcare, USA) was used for affinity chromatography according to the manufacturer's instructions. The purified proteins were concentrated, and the buffer was exchanged for 50 mM Tris–HCl buffer (pH 7.4) for enzymatic assays.

## Activity assays, biochemical properties, and kinetic analysis of PnGH1

A reaction mixture that contained 50 mM Tris–HCl, pH 5.0, 500 μM ginsenosides (**1**–**4**, **6**–**10**) as substrates, and 3 μg purified protein in a final volume of 100 μL was incubated at 50 °C for 4 h. Ginsenoside Rf and baimaside were also tested as the substrate of PnGH1 in the same condition, respectively. The reactions were terminated by the addition of 300 μL of ice-cold MeOH and were centrifuged at $12,000 \times g$ for 30 min. The supernatants were analyzed using Method 4 in Supplementary Table 3.

The effects of pH value, temperature on PnGH1 activity, and the reaction time course were studied using Rb3 as the substrate. To test the optional pH value for PnGH1 activity, assays were performed in different reaction buffers in pH values ranging from 4.0 to 6.0 (citric acid–sodium citrate buffer), 6.0–8.0 ($K_2HPO_4$–$KH_2PO_4$ buffer), 7.0–9.0 (Tris–HCl buffer) and 9.0–11.0 ($Na_2CO_3$–$NaHCO_3$ buffer), respectively. The assays were incubated at different temperatures (10–80 °C) to assay the optimal reaction temperature. The time courses of the hydrolysis reaction catalyzed by PnGH1 were evaluated at different time points of 5 min–48 h. All experiments were performed in triplicate.

To determine the kinetic values ($K_M$, $k_{cat}$, and $k_{cat}/K_M$) of PnGH1 towards Rb3, reactions were performed in a total volume of 200 μL containing 1 μg purified protein, 50 mM Tris–HCl buffer (pH 5.0) and varying concentrations (30–8000 μM) of Rb3. The reactions were conducted at 60 °C for 30 min. Supernatants were analyzed by analytical reverse-phase HPLC using Method 2 (Supplementary Table 3). The quantitative determination of product Fd was calculated based on the standard curve of the reference compound. Data fitting was performed using the Michaelis–Menten equation in GraphPad Prism 7.

## Transient expression of *PnGH1* in *N. benthamiana*

The coding region of *PnGH1* was cloned into pCAMBIA1300-35s binary vector with *Sac* I and *Xba* I restriction digestion sites using primers in Supplementary Table 4. The plasmid was then transformed into *Agrobacterium tumefaciens* EHA105. Subsequent agroinfiltration procedures were the same as reported before[40]. Leaves were harvested after 2 and 48 h of the injection of Rb3 (200 μM) and ultrasonically extracted in water as described above. The obtained solution was centrifuged, and the supernatants were analyzed by HPLC–MS.

## Structure modeling, molecular docking, and dynamic simulation analysis of PnGH1

Structure models of PnGH1 were constructed by AlphaFold2[41], Swiss-Model database[42], and MODELER software v.9.14[43], respectively. The established models were further evaluated by SAVES tools[44]. The overall structure of the obtained three models showed high similarity except for some random coil regions (Supplementary Fig. 16a). Whereas the AlphaFold2 constructed model exhibited low prediction accuracy around residues 360–400 (Supplementary Fig. 16b–d) of which some residues are potentially involved in the hydrolysis process. Thus, based on comprehensive comparison and evaluation (Supplementary Figs. 16 and 17), the homology modeling structure constructed by MODELER software was selected for subsequent analyses. Molecular docking studies were carried out in AutoDock Vina[45]. Os4BGlu12, a β-glucosidase (PDBID 3PTK) identified from *Oryza sativa* L. with 44.74% sequence identity to PnGH1 was used as the template

structure for substrate binding pocket search. The ligands were docked into the active site using the grid box dimension $40 \times 40 \times 40$ Å in *x*, *y*, and *z* with a grid spacing of 0.375 Å via a Lamarkian genetic algorithm methodology. Molecular dynamics (MD) simulation was applied using the Amber20 software package as reported previously[46]. All residues were modeled in their standard protonation states where E232 and E425 were set as neutral. Pre-simulation was performed for 10 ns with a 3.5 Å distance restraint between the OE2 atom of the carboxylate group of E425 and the anomeric carbon of Rg3 before the subsequent 50 ns dynamic simulations.

## Mutant construction and enzymatic assays

The mutants of PnGH1 were obtained using the Fast Mutagenesis system (Transgen Biotech, Beijing, China). Primers used are shown in Supplementary Table 4. The heterologous expression, protein purification, and enzyme assay procedures were the same as described above.

## Subcellular localization of PnGH1

GFP-fused constructs were prepared by inserting *PnGH1* into pCAMBIA1300-GFP vector. The sequence-verified construct was then transformed into *A. tumefaciens* strain through electroporation. The recombinant *A. tumefaciens* strain was grown at 30 °C in liquid YEB media containing 50 μg mL$^{-1}$ kanamycin. For inoculation, bacterial cells were resuspended in 10 mM MgCl₂ with the $OD_{600}$ value of 0.6. The transient expression of GFP fusion proteins was performed in *Arabidopsis* protoplasts.

## Fungal strains

Two foliar pathogenic fungi, i.e., *M. acerina* (SDM5) and *A. panax* (LCHB01), and two non-foliar pathogenic fungi, i.e., *F. oxysporum* (LCGF3), and *C. orchidophilum* (LCTJ-02), were used in this study. These fungal strains were previously isolated and purified from the susceptible *P. notoginseng* leaves or roots cultivated in Pu'er City, Yunnan Province, China, and identified by the State Key Laboratory for Conservation and Utilization of Bio-Resources in Yunnan, Yunnan Agricultural University. All fungi were maintained during the experiments on a potato dextrose agar (PDA) plate at either 20 or 25 °C to obtain vegetative growth (20 °C for *M. acerina* and *A. panax*, 25 °C for *F. oxysporum* and *C. orchidophilum*).

## Plant inoculation

A mycelial disc (5 mm diameter) was punched from the actively growing edge of the fungal PDA plates. For in vivo inoculation, one-year-old *P. notoginseng* plants were inoculated separately with *M. acerina*, *A. panax*, *F. oxysporum* and *C. orchidophilum* by placing the mycelial disc on the surface of leaves and cultured at 20 °C and 70% relative humidity. *P. notoginseng* inoculated with PDA disc without fungus served as a control. After 5- or 10-day post-inoculation (DPI), different leaf zones (Zone I–III) from *P. notoginseng* plants were collected and immediately flash-frozen for the chemical analysis with 8–10 biological replicates. For a time-course study, one-year-old *P. notoginseng* was inoculated with *M. acerina*, and three zones of leaves were harvested at different time points (0.5, 1, 3, 5, 8, 11, 14, and 17 days) after the inoculation for chemical analysis with 4–8 biological replicates.

## Antifungal activity assay in vitro

In vitro antifungal activity of total saponins or individual ginsenosides against *M. acerina*, *A. panax* and *F. oxysporum* was measured by the mycelium linear growth rate method[47]. Briefly, the total saponins or ginsenosides were dissolved in DMSO to a series of stock solutions and sterilized by a sterile syringe PVDF filter with a 0.22 μm pore size. The DMSO stock solution was diluted 2.5:1000 (v/v) in the sterilized PDA at 50–60 °C to make mediums with different drug concentrations. Equal

amounts of DMSO without ginsenoside in PDA were prepared as a control group. A mycelial disc (5 mm) was punched from the actively growing edge of the fungal PDA plates and inoculated in the center of PDA plates containing different concentrations of tested ginsenosides. Each colony diameter was measured 5 days after inoculation by the cross-bracketing method[48]. The growth inhibition rate was calculated as the following equation:

$$I(\%) = (C - T)/(C - d) \times 100.$$

where $I$ is the inhibition rate (%), $d$ is the diameter of the mycelium disc (5 mm). $C$ and $T$ are the average colony diameters of the mycelium of the control and treatment, respectively. The experiment was individually repeated at least thrice.

### Antifungal activity assay in vivo
The control efficacy of total saponins and Rb3/Fd against *M. acerina* in the leaves of *P. notoginseng* was evaluated by pot experiments[49]. The 4-week-old PNL were disinfected with sterilized water and dried naturally. The refined total saponins or individual ginsenosides were separately dissolved in DMSO and diluted 2.5:1000 (v/v) with the sterilized water containing 0.1% tween 80 as a surfactant to yield a series of final concentrations (2, 3, and 4 mg mL$^{-1}$ for total saponins, 0.5, 1 and 2 mM for the compounds). Carbendazim (200 μM) and blank solution were used as the positive and negative control, respectively. These solutions were sprayed on the surface of PNL 3 times per day for 3 consecutive days. On the 4th day, the mycelial plugs (1 mm diameter) of *M. acerina* were inoculated into the previously processed PNL. Then, the plants were carefully placed in a transparent chamber with 70% relative humidity at 20 °C under a light/dark cycle (12/12 h). After 6 days, the infected leaves were scanned by a Scanner (Perfection V850 Pro, Epson, Suwa, Japan) supported by the SilverFast software. The lesion area was calculated using Adobe Photoshop software[50]. Each treatment had 10–15 biological replicates, and the experiment was individually repeated thrice.

### Transmission electron microscope (TEM) observations
TEM observations on the subcellular organelle states of PNL were performed by the protocols[51]. The leaf samples were fixed with 2.5% glutaraldehyde in PBS (0.1 M, pH 7.2) for 24 h at 4 °C and further fixed in 1% OsO$_4$ aqueous solution for 2 h. After washing with PBS thrice, the specimens were dehydrated by a gradient ethanol series, and embedded in dry capsules filled with pure White's resins (London Resin Company Ltd.) and polymerized at 70 °C overnight. The sections (70 nm thickness) were cut using an ultramicrotome (Leica EM UC7, Leica, Germany) and stained using uranyl acetate and lead citrate. The stained sections were observed and captured by a transmission electron microscope (Hitachi 7650, Tokyo, Japan). The experiment was performed in triplicate.

### Extraction of extracellular crude enzymes
*M. acerina*-secreted exoenzymes induced by PNL in vivo were prepared as the method described previously with minor modification[52]. To eliminate the possible influence of the active enzymes that existed in the leaves, the proteins of PNL were inactivated by continuous steaming for 30 min. The steamed leaves were inoculated with active *M. acerina* disc (5 mm diameter) and cultured at 20 °C and 70% relative humidity. The steamed leaves without fungal inoculation were used as in vivo control. After 7 days post-inoculation, the mycelium over the lesion was removed, and 1 g of decayed leaves were collected and extracted by 1.5 mL of extraction buffer (1 mol L$^{-1}$ NaCl, 20 mmol L$^{-1}$ Tris–HCl, pH 7.4) and homogenized. After being centrifuged at 7000 × $g$ and 4 °C for 15 min, the supernatant was collected for further enrichment. The exoenzymes induced by PNL or pectin in vitro were also prepared according to the methods[53]. Briefly, an active *M. acerina*

disc (5 mm diameter) was grown in 100 mL of Czapek's Dox liquid medium. Cultures were added with 1 g pectin or healthy PNL (1 g, cutting into small pieces), and then cultured in an incubator at 150 r min$^{-1}$ and 20 °C. *M. acerina*-inoculated medium without pectin and PNL tissue was used as an in vitro control. After 11 DPI, the mycelium was removed, and the culture medium was collected and centrifuged at 7000 × $g$ for 20 min at 4 °C. The supernatant was obtained for further enrichment.

The exoenzyme was purified using a dialysis tubing with 12–14 kDa of molecular weight cut-off (MWCO) for 12 h against water at 4 °C, then concentrated by a Millipore ultrafiltration system (membrane cut-off 3 kDa). After being centrifuged, the clarified supernatant was collected, and their total proteins were quantified by a Pierce™ BCA Protein Assay Kit (Thermo Fisher Scientific Inc., Rockford, IL, USA). The pectin methylgalacturonase (PMG) activity was determined by a commercial PMG assay kit (Grace Biotech, Suzhou, China) according to the manufacturer's instructions and normalized to the total protein (mg).

### Exoenzyme inoculation
The sterile absorbent cotton containing the exoenzymes induced by PNL in vivo and in vitro, and pectin (3 and 8 mg mL$^{-1}$) was placed on a 4-week-old PNL that was pre-wounded by a fine needle. 3–4 leaves of each seedling were inoculated. Leaves treated with sterile water were used as blank control. After 4 days' culture at 20 °C and 70% relative humidity, the lesion area of leaves was photographed and observed by TEM, and their chemical compositions were analyzed by HPLC-UV as described above. Each treatment had 6–9 biological replicates, and the experiment was individually repeated thrice.

### Cloning, heterologous expression, and functional identification of homologous genes *PgGH1*, *PgGH2* from *P. ginseng*, and *PqGH1* from *P. quinquefolium*
Homologous glucosidase genes were blasted in the *P. ginseng* genome database, which is published on http://www.herbgenome.com/, and the transcriptome database of *P. quinquefolium*, which is sequenced by the BGISEQ-500 sequencing platform (BGI, Wuhan, China), respectively, using *PnGH1* as a search template. The specific primers used for the cloning of *PgGH1*, *PgGH2*, and *PqGH1* genes are shown in Supplementary Table 4. The following heterologous expression and functional identification procedures were the same as *PnGH1*, as described above.

### Statistical analysis
All numerical data are expressed as mean ± standard deviation (SD) and analyzed using Student's $t$-test and one-way analysis of variance (ANOVA) with Tukey's post hoc test. A $p$-value of <0.05 was considered to be statistically significant.

### Reporting summary
Further information on research design is available in the Nature Portfolio Reporting Summary linked to this article.

## Data availability
The sequence data for glucosidase genes generated in this study have been deposited in the National Center for Biotechnology Information (NCBI) GenBank database under accession code OQ813637 (*PnGH1*), OQ813638 (*PgGH1*), OQ813639 (*PgGH2*), and OQ813640 (*PqGH1*). The sequence data for the fungus used in this study are available in the NCBI database under accession code OP120946 (*M. acerina*), OP120959 (*A. panax*), ON081646 (*F. oxysporum*), and MZ433189 (*C. orchidophilum*). The mass spectrometry proteomics data of the tryptic peptides have been deposited to the ProteomeXchange Consortium *via* the PRIDE[54] partner repository with the dataset identifier PXD048221. Source data are provided with this paper.

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

## Acknowledgements
This work was financially supported by grants from the Research Committee of the University of Macau (MYRG2022-00020-ICMS, J.-B.W.), and the Science and Technology Development Fund, Macau SAR (File no. 0052/2022/A1, J.-B.W. and 005/2023/SKL). We also acknowledge the funding support from the Open Research Project Program of the State Key Laboratory of Quality Research in Chinese Medicine (University of Macau) (No. SKL-QRCM-OP23012, S.-P.S.), and the Open Fund of State Key Laboratory of Bioactive Substance and Function of Natural Medicines (No. GTZK202203, S.-P.S.). This work was also supported by funding support from MOF and MARA of China Agriculture Research System (No. CARS-21, X.H.), and Major Science and Technology Project of Yunnan (No. 202204BI090003 and No. 202205AF150018, X.H.). We thank Shuai Zhang, Huiling Wang, Jiaqing Wu, and Yahui Yang (Yunnan Agricultural University) for their assistance in greenhouse experiments. We also thank Chris Koon Ho Wong, Zhiqiang Dong, and Qing Lan from the Faculty of Health Sciences, University of Macau, for their support in antifungal experiments.

## Author contributions
J.-B.W. and L.-J.M. conceived the project; L.-J.M., J.-B.W., S.-P.S. and X.H. designed the studies; L.-J.M. performed open field and greenhouse experiments with the help of L.G., J.C., F.L., N.M. and K.Y.; X.L., Y.L., B.Z., X.C. and S.-P.S. identified PnGH1 and performed the biochemistry work of PnGH1. F.-Q.Y. provided intellectual contribution. L.-J.M., J.-B.W., X.L. and S.-P.S. prepared the manuscript. J.-B.W., S.-P.S. and X.H. acquired funding.

## Competing interests
The authors declare no competing interests.
