## [Peer Review File · Nature Communications]

REVIEWER COMMENTS

Reviewer #1 (Remarks to the Author):

Ma et al. discovered a β -glucosidase in *Panax* species that generates ginsenosides with improved antifungal activities. They first found that ginsenosides are hydrolyzed in the lesion area of *P. notoginseng* leaves with pathogenic fungal infection. The responsible β -glucosidase was identified by fractional purification and transcriptional analysis. The inhibitory activities of the hydrolyzed products were also analyzed. The manuscript is well-written and describes an important chemical defense system in *Panax notoginseng*, and I believe this work will be of highly interest for the community of plant natural product chemistry. Therefore, I recommend accepting this manuscript for publication in the journal. I have also listed minor questions for improvement.

1. Is there any correlation between the expressions of the genes for the ginsenoside biosynthesis and β -glucosidase when infected by fungi?
2. What is the estimated concentration of Hydrolysis products in plants? Is it higher than the effective concentration determined by in vitro analysis?
3. Can the authors exclude the possibility that metabolites of *M. acerina* activate the two-component system?

Reviewer #2 (Remarks to the Author):

The paper by Ma et al is a comprehensive description of a two-component system in *Panax* species. It shows that *Panax* species accumulate saponin that are specifically hydrolysed by a beta-glucosidase located in chloroplast. Upon pathogenic fungal infection the chloroplast is hydrolyzed, the beta-glucosidase is released and transforms the stored saponin into more bioactive forms.

This is indeed an interesting study, but the novelty is restricted to the *Panax* system as a saponin two-component system was recently described by Lacchini et al, albeit with a nuclear located beta-glucosidase. The major shortcoming of the paper is thus that it is focused entirely on the *Panax* system and fails to include state-of-the-art of other two-component systems regarding saponins nor to discuss the *Panax* system in regards to other beta-glucosidases in other two-component systems. Eg the take-

home message of phylogenetic analysis in supplementary figure 15 is not clear, nor how the tree was constructed. Perhaps focusing on a tree of plant beta-glucosidases and then include species names and substrate types would be more informative as a basis for an evolutionary perspective. I tried to locate the relevant beta-glucosidase from the Lacchini paper, but could not find it, as it requires that you know the accession member.

The authors show that tissue damage also cleaves the saponins, but then they write that insects do not induce cleavage. It is not clear if the authors are discussing chewing or sucking insects nor how they did the experiments and what was actually analyzed. I would assume that chewing insects would destroy chloroplasts and release beta-glucosidases, at least there are examples of this in the literature. Normally the regurgitate is analyzed and not the leaf tissue that was taken a bite from.

Standard deviations for K_m and K_{cat} should be rounded up/down.

The authors discuss rates of turnover, but a rate would imply a time component, perhaps just calling it turnover in figure 1 would be more appropriate.

Some of the figures are very illustrative and helpful, but 3a is perhaps too generic. There are also numerous figures in the supplementary that are not referred to in the main text and that could be omitted.

Supplementary figure 20 would be relevant to include in the main text.

All in all, the readability and relevance for a broader audience of the paper could be increased by using less abbreviations and by writing the paper in a way that non-Panax specialist can more easily follow the paper and where the state-of-the-art is lifted from a Panax focus.

Reviewer #3 (Remarks to the Author):

The paper by Ma et al. describes a two-component chemical defense system in Panax species, and demonstrated that a possible route for disease management in Panax species by developing botanical pesticides. This appears to be an interesting study, which is systematically and nicely presented.

Generally, the data are described in a logic way and the manuscript is well written. However, to improve clarity of this submission, there are some points which need to take into consideration.

Regarding the specificity of PnGH1

1. Some fungi also have β -glucosidase, which has been proved to be able to catalyze the hydrolysis of ginsenosides. Have the authors identified whether the foliar pathogenic fungi used in this study contained β -glucosidases or not?
2. Three β -glucosidases, PnGH1, PnGH1-2 and PnGH-3, have been identified in *P. notoginseng* leaves. Have the authors confirmed the recombinant expression of PnGH1, PnGH1-2 and PnGH-3 by western blot? Besides, Supplementary Fig.4 only showed the *E. coli* cells lysates before and after induction. The purification data of GH2 and 3 were missing.
3. In Fig.3a and Fig.3b, the SDS-PAGE showed two different molecular weight bands of PnGH1, please explain why. In addition, as PnGH1 is a 546-amino-acid protein, the SDS-PAGE results in both Fig. 3a and 3b were not consistent with its theoretical molecular weight.
4. The authors claimed that GH1 was a novel β -glucosidase with high enzyme stability, catalytic efficiency and hydrolytic regiospecificity. Have the authors compared GH1 with β -glucosidases in other species?

Regarding the antifungal activity of hydrolysis products

1. The efficacy control for hydrolytic products as well as carbendazim were compared at different dosage levels, which were 4 mg/mL (or 1mM) for hydrolytic products and 200 μ M for carbendazim. As the dosage is very important for evaluating efficacy, the conclusion the authors made is not very reliable.
2. Is it possible that the hydrolysis products will also target beneficial fungi? Please provide experimental evidences.

Other minor comments

1. The "Abstract" does not briefly address the contents of this study.

2. PnGH1 should be written as PnGH1.

Response Notes

Reviewer #1:

*Overall: Ma et al. discovered a β -glucosidase in Panax species that generates ginsenosides with improved antifungal activities. They first found that ginsenosides are hydrolyzed in the lesion area of *P. notoginseng* leaves with pathogenic fungal infection. The responsible β -glucosidase was identified by fractional purification and transcriptional analysis. The inhibitory activities of the hydrolyzed products were also analyzed. The manuscript is well-written and describes an important chemical defense system in *Panax notoginseng*, and I believe this work will be of highly interest for the community of plant natural product chemistry. Therefore, I recommend accepting this manuscript for publication in the journal. I have also listed minor questions for improvement.*

- We sincerely appreciate the reviewer's positive and encouraging comments, which were of great help in revising the manuscript.
- We have addressed all the comments and concerns and revised the manuscript accordingly.

1. *Is there any correlation between the expressions of the genes for the ginsenoside biosynthesis and β -glucosidase when infected by fungi?*

- We sincerely appreciate the reviewer for the valuable and professional comments.
- To address this question, we collected the *P. notoginseng* leaves after being infected by the necrotrophic fungus *M. acerina* at different time points, including 0 h (control), 6 h, 12 h, 24 h, 48 h and 72 h. These samples were submitted for comparative transcriptome analysis. Key genes involved in the biosynthesis of ginsenosides in *P. notoginseng* (Fig. R1a) were searched through homology blast based on previously reported sequences (Luo *et al.*, 2011; Niu *et al.*, 2014; Xu *et al.*, 2018; Jiang *et al.*, 2022; Li *et al.*, 2022), including HMG-CoA reductase (*HMGR*, Accession No. KJ578757.1), farnesyl diphosphate synthase (*FPPS*, Accession No. KC953034.1), squalene synthase (*SQS*, Accession No. HQ456918.1, KC953032), squalene epoxidase (*SQE*, Accession No. AB122078.1, AFV92748.), dammarenediol-II synthase (*DS*, Accession No. KC953035), cytochrome P450 (*P450*, Accession No. MZ288741) and UDP-glycosyltransferase (*UGT*, Accession No. QOJ43864.1, QOJ43866.1). The expression profile of these genes and *PnGHI* was carefully analyzed by constructing a z-Score heatmap (Fig. R1b) based on their FPKM values. As shown in Fig. R1b, key genes involved in ginsenoside biosynthesis were consistently upregulated after being infected by *M. acerina*, and most of them reached the highest expression level at 24 h. *PnGHI* also exhibited an enhanced expression pattern in three samples (6 h-3, 12 h-3 and 24 h-2), with FPKM values almost six (6 h-3 and 12 h-3) and three times (24 h-2) higher compared to the control. These observations

suggested that the biosynthesis of ginsenosides and *PnGHI* gene were closely related to the chemical defence of *P. notoginseng* against *M. acerina* infection. These findings were added to the discussion section and Supplementary Fig. 22 in the revised manuscript. Besides, it was also noticed that after infection for 24 h, these genes were downregulated gradually to the normal level. The mechanism underlying this phenomenon is unknown. Supposedly, this may be physiologically required for specific homeostasis processes in plants.

Fig. R1. (a) Biosynthesis of PPD-type ginsenosides in *P. notoginseng* (ginsenoside Rg3 as an example). Abbreviations: HMG-CoA, 3-hydroxy-3-methylglutaryl coenzyme A; DMAPP, dimethylallyl diphosphate; IPP, isopentenyl diphosphate; FPP, farnesyl diphosphate. (b) Expression levels of *PnGHI* and genes involved in ginsenosides biosynthesis after being infected by the necrotrophic fungus *M. acerina* at different time points. The expression values of each gene were transformed to z-scores based on the FPKM values and shown as a heatmap using Origin 2019 software.

2. What is the estimated concentration of hydrolysis products in plants? Is it higher than the effective concentration determined by *in vitro* analysis?

- The levels of the hydrolysis products in decayed tissue gradually increased along with infection duration. According to the raw data of Fig. 2h and Fig. S2, the absolute contents of hydrolysis products, including gypenoside XVII (**1a**), notoginsenoside Fe (**2a**), ginsenoside Rd2 (**3a**), and notoginsenoside Fd (**4a**), in decayed tissue (wet weight) at 5 DPI and 11 DPI are showed in the table below. The total content of these four hydrolysis products is also calculated.

	Gyp XVII (1a, mg/g)	Fe (2a, mg/g)	Rd2 (3a, mg/g)	Fd (4a, mg/g)	Total (mg/g)
5 DPI	1.49 ± 0.47	6.28 ± 2.05	3.38 ± 1.03	37.01 ± 4.74	48.36 ± 5.07
11 DPI	1.91 ± 0.61	16.63 ± 5.96	9.53 ± 3.11	87.15 ± 9.81	114.77 ± 12.06

- The total concentration of hydrolysis products at 5 DPI and 11 DPI are 48.36 ± 5.07 mg/g and 114.77 ± 12.06 mg/g of decayed leaves (wet weight), respectively. The hydrolysis products (TS-a) at 4 mg/mL exhibited obvious inhibitory effects against *M. acerina* growth *in vitro* and *in vivo*. Considering the density of the broth culture medium (0.61 g/mL), 4 mg/mL is equal to 6.56 mg/g of the medium. Thus, the concentration of hydrolysis products in decayed tissue is much higher than the effective concentration determined by *in vitro* test.

3. Can the authors exclude the possibility that metabolites of *M. acerina* activate the two-component system?

- We sincerely appreciate the reviewer for the careful review and valuable comments.
- In our study, the crude exoenzymes of *M. acerina* whose expression was induced by exposure to PNL tissue (both *in vivo* and *in vitro*) or pectin can activate the two-component system, rather than the broth culture of *M. acerina*.
- As shown in Figs. 4a, 4c and 4f, *M. acerina*-inoculated medium without pectin or PNL tissue serves as *in vitro* control (Con_vitro), Con_vitro failed to produce any lesions on PNLs and trigger ginsenoside hydrolysis. Collectively, only induced exoenzymes isolated from *M. acerina* can activate the two-component system.
- We also exclude the possibility that *M. acerina* triggers ginsenoside hydrolysis. As shown in Fig. 2f, to determine whether a broth culture of a pathogenic fungus could cause the ginsenoside hydrolysis, Rb3 was spiked into an *M. acerina* culture, and the spiked culture was incubated for 11 days. The hydrolysis product Fd was not detected, and no significant difference in Rb3 content was observed at different intervals, indicating that enzymes secreted by *M. acerina* cannot trigger Rb3 hydrolysis.

Reviewer #2:

1. *This is indeed an interesting study, but the novelty is restricted to the Panax system as a saponin two-component system was recently described by Lacchini et al, albeit with a nuclear located beta-glucosidase. The major shortcoming of the paper is thus that it is focused entirely on the Panax system and fails to include state-of-the-art of other two-component systems regarding saponins nor to discuss the Panax system in regards to other beta-glucosidases in other two-component systems.*
 - We sincerely appreciate the reviewer for the careful review and valuable comments.
 - Recently, Lacchini *et al.* reported the incredible and gorgeous work that *Medicago truncatula*, a model legume, has evolved a two-component system composed of a nucleolar-localized β -glucosidase (G1) and triterpene saponins, acting as a saponin bomb. In this study, we first described a two-component chemical defence system involving chloroplast-localized β -glucosidase and 20(*S*)-protopanaxadiol ginsenosides in *Panax* species. Lacchini's paper and our study adequately revealed that the saponin bomb model might be a widespread two-component system in many plant species. Additionally, *Panax* species encompasses several important ginseng species with high medicinal and economic values, including *Panax notoginseng* (Notoginseng), *Panax ginseng* (Asian Ginseng) and *Panax quinquefolium* (American Ginseng). Our findings are of great significance in revealing a chemical defence against fungal infection in *Panax* species and in developing botanical pesticides for disease management in *Panax* species, the highly valued medicinal plants.
 - As suggested, the state-of-the-art of two-component systems involving triterpene saponins has been included in the revised manuscript.
2. *Eg the take-home message of phylogenic analysis in supplementary figure 15 is not clear, nor how the tree was constructed. Perhaps focusing on a tree of plant beta-glucosidases and then include species names and substrate types would be more informative as a basis for an evolutionary perspective. I tried to locate the relevant beta-glucosidase from the Lacchini paper, but could not find it, as it requires that you know the accession member.*
 - We sincerely appreciate the reviewer for the careful review and constructive comments.
 - As suggested, we have reconstructed the phylogenic tree as shown in Supplementary Fig. 15b. The take-home message of the phylogenic analysis was noted in the figure legends. The new tree was mainly focused on plant-originated β -glucosidases involved in plant defence. The listed sequences were selected from Lacchini's paper (Lacchini *et al.*, 2023) and Morant's paper (Morant *et al.*, 2008a). Species names and accession numbers of each sequence were provided in the legend. The substrate type of each sequence was displayed beside the enzyme on the right side, respectively.

- As reported before, currently identified β -glucosidases involved in plant defence from monocots and dicots are separated, indicating an independent evolutionary development. Their substrate types mainly include cyanogenic glucosides, isoflavonoid glucosides, glucosinolates, etc. Only MtG1 was reported to hydrolyze triterpene saponins (3-Glc-28-Glc-medicagenic acid) in *Medicago truncatula*. However, it is phylogenetically located in the clade of isoflavone conjugate-hydrolyzing β -glucosidase and many cyanogenic glucoside glucosidases. Interestingly, β -glucosidases identified from *Panax* species in this paper, which also accepted triterpenoid saponins as substrate, were phylogenetically close to neither MtG1 nor other β -glucosidases. The four glucosidases from *Panax* species were clustered as a new subclade, indicating a distinct evolutionary branch.
3. *The authors show that tissue damage also cleaves the saponins, but then they write that insects do not induce cleavage. It is not clear if the authors are discussing chewing or sucking insects nor how they did the experiments and what was actually analyzed. I would assume that chewing insects would destroy chloroplasts and release beta-glucosidases, at least there are examples of this in the literature. Normally the regurgitate is analyzed and not the leaf tissue that was taken a bite from.*
- Thank you very much for your professional and valuable comments that helped us improve this manuscript.
 - We totally agree with the reviewer that chewing insects may destroy chloroplasts and activate the two-component system. We also agree to analyze regurgitate, rather than leaf tissue bitten by insects, to identify whether ginsenoside hydrolysis is associated with insect attack. Unfortunately, in this study, the leaves bitten by herbivores were collected from open fields and further analyzed. We didn't further identify which type of insects are, chewing or sucking, due to the expertise limitation. Thus, the context regarding insect attacks has been removed from the revised manuscript to avoid confusing the readers.
4. *Standard deviations for Km and Kcat should be rounded up/down.*
- As suggested, we have made revisions accordingly in the main text and figures.
5. *The authors discuss rates of turnover, but a rate would implies a time component, perhaps just calling it turnover in figure 1 would be more appropriate.*
- We fully agree with the reviewer that "hydrolysis turnover" is more appropriate than "hydrolysis rate" in Figure 1. We have made revisions accordingly throughout the whole manuscript.
6. *Some of the figures are very illustrative and helpful, but 3a is perhaps too generic. There are also numerous figures in the supplementary that are not referred to in the main text and that could be omitted.*

- We sincerely appreciate the reviewer for the careful review and constructive comments.
 - As suggested, Figure 3a was moved from the main text to Supplementary Materials as new Supplementary Fig. 3a.
 - We have carefully reviewed the entire manuscript to eliminate the problems regarding supplementary figures.
7. *Supplementary figure 20 would be relevant to include in the main text.*
- As suggested, Supplementary Figure 20 has been moved to the main text (Fig. 4a).
8. *All in all, the readability and relevance for a broader audience of the paper could be increased by using less abbreviations and by writing the paper in a way that non-Panax specialist can more easily follow the paper and where the state-of-the-art is lifted from a Panax focus.*
- The manuscript has been improved accordingly to increase the readability and relevance for a broader audience.

Reviewer #3:

The paper by Ma et al. describes a two-component chemical defense system in Panax species, and demonstrated that a possible route for disease management in Panax species by developing botanical pesticides. This appears to be an interesting study, which is systematically and nicely presented. Generally, the data are described in a logic way and the manuscript is well written. However, to improve clarity of this submission, there are some points which need to take into consideration.

- We sincerely appreciate the reviewer's positive and encouraging comments on our manuscript.

Regarding the specificity of PnGH1

1. *Some fungi also have β -glucosidase, which has been proved to be able to catalyze the hydrolysis of ginsenosides. Have the authors identified whether the foliar pathogenic fungi used in this study contained β -glucosidases or not?*
- An increasing number of studies have demonstrated that many fungi and bacteria contain β -glucosidase that can catalyze the hydrolysis of ginsenosides.
 - Our study didn't investigate whether pathogenic fungi used in this study contain β -glucosidases. However, we exclude the possibility that *M. acerina* activates the ginsenoside hydrolysis. As shown in **Fig. 2f**, to determine whether a broth culture of pathogenic fungus could cause the ginsenoside hydrolysis, Rb3 was spiked into an *M. acerina* culture, and the

spiked culture was incubated for 11 days. The hydrolysis product Fd was not detected, and no significant difference in Rb3 content was observed at different intervals, indicating that the broth culture of *M. acerina* cannot trigger the hydrolysis of Rb3.

2. Three β -glucosidases, PnGH1, PnGH1-2 and PnGH-3, have been identified in *P. notoginseng* leaves. Have the authors confirmed the recombinant expression of PnGH1, PnGH1-2 and PnGH-3 by western blot? Besides, Supplementary Fig.4 only showed the *E. coli* cells lysates before and after induction. The purification data of GH2 and 3 were missing.

– We sincerely appreciate the reviewer for the careful review and valuable comments.

The revised Supplementary Fig. 4

- Following your suggestion, the recombinant expression of *PnGH1*, *PnGH2* and *PnGH3* has been confirmed by western blot analysis as shown in the revised Supplementary Fig. 4c. The SDS-PAGE results of the recombinant expression of *PnGH1*, *PnGH2* and *PnGH3* in *E. coli* have also been updated in the revised Supplementary Fig. 4b showing cell lysates before and after induction, as well as the purified recombinant proteins.
3. In Fig.3a and Fig.3b, the SDS-PAGE showed two different molecular weight bands of *PnGH1*, please explain why. In addition, as *PnGH1* is a 546-amino-acid protein, the SDS-PAGE results in both Fig. 3a and 3b were not consistent with its theoretical molecular weight.
- We sincerely appreciate the reviewer for the careful review and valuable comments.
 - *PnGH1* is a 546-amino-acid protein whose theoretical molecular weight is 61.85 kD.
 - In Fig. 3b (Fig. 3a in the revised paper), the recombinant *PnGH1* protein band was shown at a size larger than 70 kD. For heterologous expression, *PnGH1* gene was constructed into pET-32a vector using *EcoR* I and *Xho* I as restriction sites. Based on the vector map, *PnGH1* was expressed as a fusion protein consisting of 167 amino acids encoded from the vector at the *N*-terminal, whose theoretical molecular weight was approximately 17.9 kD. These amino acids include a Trx-tag, a His-tag and a S-tag sequences, which play crucial roles in facilitating the soluble expression of *PnGH1* in *E. coli*. Thus, the fusion protein of *PnGH1* was calculated to be around 79 kD, consistent with the observed band in Fig. 3b. We apologize for missing these details before, and the restriction sites have been noted in the experimental section in the revised version.
 - Fig. 3a (Supplementary Fig. 3a in the revised paper) represents a protein band at the size between 40–50 kD, which is lower than the theoretical molecular weight of *PnGH1*. This protein was obtained through crude enzyme extraction followed by activity-guided purification procedures. A reasonable explanation was that some of the unstable segments in the protein were cleaved during the repeated purification processes or degraded by some chemicals or *in vivo* proteases after the cell was lysed. This speculation was made based on the following observation: (1) The three-dimensional structure of *PnGH1* showed an obvious redundant region at the *N*-terminal 1-72 amino acids as shown in Fig. R2a. (2) Homology blast found that sequences in this region were not involved in the hydrolysis process. (3) A propeptide cleavage site was predicted between Lys72 and Arg73 with a score of 0.069 (Fig. R2b); (4) LC-MS proteomics analysis of the obtained protein combined with transcriptome sequence blast found *PnGH1* showing the highest matching score which is far above other candidates (Supplementary Table 1), and their sequence alignment consistency begins at Arg73 of *PnGH1* (labeled by a red triangle in Fig. R2c). To verify the speculation, we constructed the truncated *PnGH1* (73–505 aa) and achieved its soluble expression in pET-32a

vector. After removing the fusion tag through enterokinase cleavage, SDS-PAGE analysis displayed that the obtained protein represents a similar size, around ~45 kD (Fig. R2d and R2e). Most importantly, this truncated protein also demonstrated robust hydrolysis activity towards Rb3 to generate Fd (Fig.R2f). These results strongly indicated that the protein obtained from activity-guided purification in Fig. 3a (Supplementary Fig. 3a in the revised paper) probably represents the core active unit of PnGH1 and thus exhibits a relatively smaller molecular weight in SDS-PAGE.

Fig. R2. (a) The three-dimensional structure of PnGH1 with 72 residues at *N*-terminal shown in blue. (b) Propeptide cleavage site prediction using DTU Health Tech's ProP tool. (c). Sequence alignment of PnGH1 and the tryptic peptide sequences obtained through LC-MS proteomics analysis. (d-e) SDS-PAGE analysis of active protein obtained from activity-guided purification (d) and truncated PnGH1 (e). (f) HPLC analysis of the hydrolysis activity of PnGH1 and truncated PnGH1 towards Rb3.

4. The authors claimed that GH1 was a novel β -glucosidase with high enzyme stability, catalytic efficiency and hydrolytic regiospecificity. Have the authors compared GH1 with β -glucosidases in other species?

- We sincerely appreciate the reviewer for the professional suggestion.
- Regarding the stability of PnGH1, a typical $(\beta/\alpha)_8$ barrel fold, which contains eight parallel β -strands surrounded by eight helices, was observed in the three-dimensional structure of PnGH1. This structural type is a relatively stable structural fold (Czjzek *et al.*, 2001). In addition, several salt bridges were observed in PnGH1 structure, which could contribute to the stability of PnGH1 (Sansenya *et al.*, 2011). Furthermore, biochemical characterization has revealed that PnGH1 retains its activity across a wide pH range of 3.0 to 9.0 and a temperature range of 10°C to 80°C. Remarkably, more than 60% of its activity could be maintained under acidic conditions (pH value 4-5) or high temperatures (40-60 °C). Based on these observations, we described PnGH1 as a highly stable glucosidase.
- Regarding the catalytic efficiency of PnGH1 compared with other β -glucosidases, we have summarized the kinetic parameters of several reported β -glucosidases in Table R1, including β -glucosidases involved in plant defence (highlighted in blue), other plant-originated β -glucosidases hydrolyzing natural glycosides (highlighted in yellow) and microbial β -glucosidases hydrolyzing ginsenosides (highlighted in pink). From Table R1, it is easy to find that the K_M values of most reported β -glucosidases are in the range of 1–5 mM. In contrast, the K_M value of PnGH1 was determined to be 0.677 mM, indicating a relatively high substrate affinity towards ginsenoside Rb3. As for k_{cat} and k_{cat}/K_M parameters, it appears that these parameters of currently reported glucosidases varied widely. The k_{cat}/K_M values of PnGH1 ($5.8 \text{ s}^{-1}\cdot\text{mM}^{-1}$) showed no significant advantage compared with some other enzymes, such as GmICHG from *Glycine max* ($3900 \text{ s}^{-1}\cdot\text{mM}^{-1}$), SG from *Rauvolfia serpentine* ($106.4 \text{ s}^{-1}\cdot\text{mM}^{-1}$), and BsAbfA from *Bacillus subtilis* ($197.8 \text{ s}^{-1}\cdot\text{mM}^{-1}$). Accordingly, we consider PnGH1 as an efficient β -glucosidase rather than claiming it as "highly" efficient in the revised manuscript.

However, several points are worth further being noted here. Recently, Lacchini *et al.* (2023) reported a β -glucosidase MtG1 being involved in plant defence from *Medicago truncatula*, which also accepts triterpene saponins as the substrate. MtG1 was claimed as a highly efficient and fast β -glucosidase with a k_{cat}/K_M ratio of $26.299 \text{ s}^{-1}\cdot\text{mM}^{-1}$. MeLinamarase (Keresztessy *et al.*, 2001), a cyanogenic β -glucosidase involved in plant defense through cyanogenesis mechanism against small herbivores, exhibited a much lower k_{cat}/K_M value of $0.0246\pm 0.0028 \text{ s}^{-1}\cdot\text{mM}^{-1}$ toward its natural substrate linamarin. In our study, the k_{cat}/K_M value of PnGH1 ($5.8 \text{ s}^{-1}\cdot\text{mM}^{-1}$) significantly surpasses that of MeLinamarase and is comparable to that of MtG1 in terms of magnitude, suggesting that the catalytic efficiency of PnGH1 is sufficient to play a crucial role in plant defence.

Table R1. Kinetic parameters of some of the reported β -glucosidases

Glucosidase	K_M (mM)	k_{cat} (s ⁻¹)	k_{cat}/K_M (s ⁻¹ ·mM ⁻¹)	V_{max}	Species	Substrate	References
LjBGD2 LjBGD4 (mixture)	0.69±0.06			2.47±0.06 nmol/min	Lotus japonicus	Lotaustralin	Morant et al. , (2008b)
PsAH1	2.5	\	\	\	Prunus serofina Ehrh.)	Amygdalin	Kuroki and Poulton, (1986)
PsAH2	2.1	\	\	\	Prunus serofina Ehrh.)	Amygdalin	Kuroki and Poulton, (1986)
ScBxGlcGLU	2.8±0.6	169.2±25.1		2524.9±374.4 nkat/mg	Secale cereale	DIBOAGlucose	Nikus et al. , (2003)
MeLinamarase	1.05	0.0259±0.0017	0.0246±0.0028	\	Manihot esculenta Crantz	Linamarin	Keresztessy et al. , (2001)
HbLinamarase	7.6	\	\	191,000 nmol/mg ⁻¹ ·min ⁻¹	Heyea brasiliensis	Linamarin	Selmar et al. , (1987)
TaGlu1a	1.40	48.8	34.6	\	Triticum aestivum	DIBOA-Glc	Sue et al. , (2006)
TaGlu1b	1.44	214	149	\	Triticum aestivum	DIBOA-Glc	Sue et al. , (2006)
TaGlu1c	1.05	137	131	\	Triticum aestivum	DIBOA-Glc	Sue et al. , (2006)
DcBDGLU	5.78±0.20	\	\	\	Dalbergia cochinchinensis Pierre	pNPGlc	Ketudat Cairns et al. , (2000)
GmICHG	0.025±0.002	98±3	3900	\	Glycine max	Daidzein 7-O-(6''-O-malonyl- β -D-glucoside)	(Suzuki et al. , 2006)
Os3BGLu6	6.3±0.4	38.9±0.9	6.2	\	Oryza sativa L.	pNPGlc	Hua et al. , (2013)
MtG1	0.154	4.05	26.299	\	Medicago truncatula	3-Glc-28-Glc-medicagenic acid	Lacchini et al. , (2023)
MtG2	1.85	3.0	1.621	28.8 μ mol/min	Medicago truncatula	pNPGlc	Naoumkina et al. , (2007)
β -glucosidase	3.8	\	\	2500 U/mg	Olea europaea cv.	Oleuropein	Romero-Segura et al. , (2009)
Hbglu	1.32	\	\	0.73 mmol/min/L	Hevea brasiliensis	pNPGlc	Xu et al. , (2020)
Hbglu	4.38	\	\	0.11 mmol/min/L	Hevea brasiliensis	Daidzin	Xu et al. , (2020)
SG	0.09	9.8	106.4	\	Rauvolfia serpentina	Strictosidine	Barleben et al. , (2007)
bglF3	2.84±0.05	\	\	\	Flavobacterium johnsoniae	Ginsenoside Rb1	Hong et al. , (2012)
bglSk	0.167±0.003	\	\	\	Sanguibacter keddieii	Ginsenoside Rb1	Kim et al. , (2012)
bglSp	2.9±0.3	\	\	\	Sphingomonas sp.	pNPGlc	Wang et al. , (2011)
β -glucosidase	5.74	\	\	2.70 mM/h	Aspergillus sp.	Ginsenoside Rg2	Wang et al. , (2012)
β -glucosidase	9.43	\	\	2.84 mM/h	Aspergillus sp.	Ginsenoside Rf	Wang et al. , (2012)
Anxyl	1.55	\	\	\	Aspergillus niger NG1306	Ginsenoside Rb3	Zhu et al. , (2020)
BsAbfA	0.4	\	197.8	\	Bacillus subtilis	Ginsenoside Rc	Zhang et al. , (2021)
β -glucosidase	4.165	\	\	0.021 mmol/mg ⁻¹ ·min ⁻¹	Aspergillus niger	Ginsenoside Rb1	Kazan et al. , (2021)
HaGH03	0.59±0.23	\	\	\	Herpetosiphon aurantiacus	Vinaginsenoside R7	Wang et al. , (2015)
PnGH1	0.677±0.03	3.9±0.05	5.8	2.99 nmol·min ⁻¹ · μ g ⁻¹	Panax notoginseng	Ginsenoside Rb3	This paper

- Regarding the regioselectivity, we primarily referred to the high specificity of PnGH1 in recognizing β -(1,2)-glucosidic linkage. As shown in Supplementary Fig. 12, *in vitro* enzymatic assays revealed that PnGH1 could accept structurally diverse PPD-type ginsenosides, including Rb1 (1), Rc (2), Rb2 (3), Rb3 (4), Ra2 (6), Ra3 (7), Ra1 (8), Rd (9), and Rg3 (10) as substrates to produce the corresponding deglycosylated products (1a–3a and 6a–10a). These hydrolysis reactions all occurred at the β -(1,2)-linked glucosidic bond of sugar chains at the C-3 position, indicating the catalysis regioselectivity of PnGH1. PnGH1 is the first regioselective β -(1,2) glucosidase identified from *P. notoginseng*.
 - To summarize, the previous statement that "PnGH1 was a novel β -glucosidase with high enzyme stability, catalytic efficiency and hydrolytic regioselectivity" has been revised to "PnGH1 was a novel β -glucosidase with high enzyme stability and hydrolytic regioselectivity" in the discussion section.
5. *The efficacy control for hydrolytic products as well as carbendazim were compared at different dosage levels, which were 4 mg/mL (or 1mM) for hydrolytic products and 200 μ M for carbendazim. As the dosage is very important for evaluating efficacy, the conclusion the authors made is not very reliable.*
- We sincerely appreciate the reviewer for the valuable and constructive comments.
 - We agree with the review's opinion, and the relevant conclusion has been corrected accordingly. "Given that hydrolysis products show the potent antifungal activity both *in vitro* and *in vivo*, our findings also provide new insights for developing green botanical pesticides for disease management in *Panax* species."
6. *Is it possible that the hydrolysis products will also target beneficial fungi? Please provide experimental evidence.*
- We sincerely appreciate the reviewer for the careful review and valuable comments.
 - As suggested, the toxic effects of intact ginsenosides (TS-a) and their hydrolysis products (TS-b) against the growth of beneficial fungi were tested and compared *in vitro*. Two beneficial fungi, including *Acremonium sp.* D212, an endophytic fungus isolated from the buds of *Panax notoginseng*, and *Saccharomyces cerevisiae* (yeast strains WT HHY168), were used in this experiment.
 - As shown in **Fig. R1**, TS-a exhibited significantly stronger inhibitory effects against the growth of both yeast and *Acremonium sp.* D212 at the applied concentrations than TS-b, indicating that refined extracts containing hydrolysis products are more toxic to a broad spectrum of fungi, including beneficial fungi, at the tested concentration ranges than intact ginsenosides. Furthermore, compared to *M. acerina* (**Fig. 5** of the manuscript), hydrolysis products showed less inhibitory rate on *Acremonium sp.* D212 (25.2% vs 80.8% at the

concentration of 4 mg/mL).

Fig. R3 *In vitro* antifungal activity of TS-a and TS-b against (a) yeast and (b-c) *Acremonium sp.* D212. b, Inhibitory effects on colony diameter at 13 DPI and (c) corresponding growth inhibition rate.

Other minor comments

7. The "Abstract" does not briefly address the contents of this study.

- "Abstract" has been modified accordingly. Due to the 200-word limit, a general introduction and non-technical summary of the main results and their implications have been included.

8. *PnGHI* should be written as *PnGHI*.

- As suggested, the correction has been made accordingly. "*PnGHI*" referred to genes were all in italics, while "PnGHI" represents proteins were all in normal representation.

References

- Barleben L, Panjikar S, Ruppert M, Koepke J, Stöckigt J (2007) Molecular architecture of strictosidine glucosidase: the gateway to the biosynthesis of the monoterpene indole alkaloid family. *Plant Cell* **19**: 2886 – 2897
- Czjzek M, Cicek M, Zamboni V, Burmeister WP, Bevan DR, Henrissat B, Esen A (2001) Crystal structure of a monocotyledon (maize ZMGlu1) beta-glucosidase and a model of its complex with p-nitrophenyl beta-D-thioglucoside. *Biochem J* **354**: 37 – 46
- Hong H, Cui CH, Kim JK, Jin FX, Kim SC, Im WT (2012) Enzymatic biotransformation of ginsenoside Rb1 and gypenoside XVII into ginsenosides Rd and F2 by recombinant β -glucosidase from *Flavobacterium johnsoniae*. *J Ginseng Res* **36**: 418 – 424
- Hua Y, Sansenya S, Saetang C, Wakuta S, Ketudat Cairns JR (2013) Enzymatic and structural characterization of hydrolysis of gibberellin A4 glucosyl ester by a rice β -D-glucosidase. *Arch Biochem Biophys* **537**: 39 – 48
- Jiang Z, Gao H, Liu R, Xia M, Lu Y, Wang J, Chen X, Zhang Y, Li D, Tong Y (2022) Key glycosyltransferase genes of *Panax notoginseng*: identification and engineering yeast construction of rare ginsenosides. *ACS Synth Biol* **11**: 2394 – 2404
- Kazan A, Hu X, Stahl A, Frerichs H, Smirnova I, Yesil-Celiktas O (2021) An enzyme immobilized microreactor for continuous – flow biocatalysis of ginsenoside Rb1. *J Chem Technol Biot* **96**: 3349 – 3357
- Keresztessy Z, Brown K, Dunn MA, Hughes MA (2001) Identification of essential active-site residues in the cyanogenic beta-glucosidase (linamarase) from cassava (*Manihot esculenta* Crantz) by site-directed mutagenesis. *Biochem J* **353**: 199 – 205
- Ketudat Cairns JR, Champattanachai V, Srisomsap C, Wittman-Liebold B, Thiede B, Svasti J (2000) Sequence and expression of Thai Rosewood beta-glucosidase/beta-fucosidase, a family 1 glycosyl hydrolase glycoprotein. *J Biochem* **128**: 999 – 1008
- Kim JK, Cui CH, Yoon MH, Kim SC, Im WT (2012) Bioconversion of major ginsenosides Rg1 to minor ginsenoside F1 using novel recombinant ginsenoside hydrolyzing glycosidase cloned from *Sanguibacter keddiei* and enzyme characterization. *J Biotechnol* **161**: 294 – 301
- Kuroki GW, Poulton JE (1986) Comparison of kinetic and molecular properties of two forms of amygdalin hydrolase from black cherry (*Prunus serotina* Ehrh.) seeds. *Arch Biochem Biophys* **247**: 433 – 439
- Lacchini E, Erffelinck ML, Mertens J, Marcou S, Molina-Hidalgo FJ, Tzfadia O, Venegas-Molina J, Cárdenas PD, Pollier J, Tava A, Bak S, Höfte M, Goossens A (2023) The saponin bomb: a nucleolar-localized β -glucosidase hydrolyzes triterpene saponins in *Medicago truncatula*. *New Phytol* **239**: 705 – 719

- Li Y, Li J, Diao M, Peng L, Huang S, Xie N (2022) Characterization of a group of UDP-glycosyltransferases involved in the biosynthesis of triterpenoid saponins of *Panax notoginseng*. *ACS Synth Biol* **11**: 770 – 779
- Luo H, Sun C, Sun Y, Wu Q, Li Y, Song J, Niu Y, Cheng X, Xu H, Li C, Liu J, Steinmetz A, Chen S (2011) Analysis of the transcriptome of *Panax notoginseng* root uncovers putative triterpene saponin-biosynthetic genes and genetic markers. *BMC Genomics* **12 Suppl 5**: S5
- Morant AV, Jørgensen K, Jørgensen C, Paquette SM, Sánchez-Pérez R, Møller BL, Bak S (2008a) beta-Glucosidases as detonators of plant chemical defense. *Phytochemistry* **69**: 1795 – 1813
- Morant AV, Bjarnholt N, Kragh ME, Kjaergaard CH, Jørgensen K, Paquette SM, Piotrowski M, Imberty A, Olsen CE, Møller BL, Bak S (2008b) The beta-glucosidases responsible for bioactivation of hydroxynitrile glucosides in *Lotus japonicus*. *Plant Physiol* **147**: 1072 – 1091
- Naoumkina M, Farag MA, Sumner LW, Tang Y, Liu CJ, Dixon RA (2007) Different mechanisms for phytoalexin induction by pathogen and wound signals in *Medicago truncatula*. *Proc Natl Acad Sci U S A* **104**: 17909 – 17915
- Nikus J, Esen A, Jonsson LM (2003) Cloning of a plastidic rye (*Secale cereale*) β -glucosidase cDNA and its expression in *Escherichia coli*. *Physio. Plantarum* **118**: 337 – 345
- Niu Y, Luo H, Sun C, Yang TJ, Dong L, Huang L, Chen S (2014) Expression profiling of the triterpene saponin biosynthesis genes FPS, SS, SE, and DS in the medicinal plant *Panax notoginseng*. *Gene* **533**: 295 – 303
- Romero-Segura C, Sanz C, Perez AG (2009) Purification and characterization of an olive fruit beta-glucosidase involved in the biosynthesis of virgin olive oil phenolics. *J Agric Food Chem* **57**: 7983 – 7988
- Sansanya S, Opasiri R, Kuaprasert B, Chen C-J, Cairns JRK (2011) The crystal structure of rice (*Oryza sativa* L.) Os4BGlu12, an oligosaccharide and tuberonic acid glucoside-hydrolyzing β -glucosidase with significant thioglucohydrolase activity. *Arch Biochem Biophys* **510**: 62 – 72
- Selmar D, Lieberei R, Biehl BI, Voigt Jr (1987) Hevea linamarase – a nonspecific β -glycosidase. *Plant Physiol* **83**: 557 – 563
- Sue M, Yamazaki K, Yajima S, Nomura T, Matsukawa T, Iwamura H, Miyamoto T (2006) Molecular and structural characterization of hexameric beta-D-glucosidases in wheat and rye. *Plant Physiol* **141**: 1237 – 1247
- Suzuki H, Takahashi S, Watanabe R, Fukushima Y, Fujita N, Noguchi A, Yokoyama R, Nishitani K, Nishino T, Nakayama T (2006) An isoflavone conjugate-hydrolyzing beta-glucosidase from the roots of soybean (*Glycine max*) seedlings: purification, gene cloning, phylogenetics, and cellular localization. *J Biol Chem* **281**: 30251 – 30259
- Wang D, Yu H, Song J, Xu Y, Jin F (2012) Enzyme kinetics of ginsenosidase type IV hydrolyzing 6-O-multi-glycosides of protopanaxatriol type ginsenosides. *Process Biochem* **47**: 133 – 138

- Wang L, Liu QM, Sung BH, An DS, Lee HG, Kim SG, Kim SC, Lee ST, Im WT (2011) Bioconversion of ginsenosides Rb(1), Rb(2), Rc and Rd by novel β -glucosidase hydrolyzing outer 3-O glycoside from *Sphingomonas* sp. 2F2: cloning, expression, and enzyme characterization. *J Biotechnol* **156**: 125 - 133
- Wang RF, Zheng MM, Cao YD, Li H, Li CX, Xu JH, Wang ZT (2015) Enzymatic transformation of vana-ginsenoside R 7 to rare notoginsenoside ST-4 using a new recombinant glycoside hydrolase from *Herpetosiphon aurantiacus*. *Appl Microbiol Biot* **99**: 3433 - 3442
- Xu J, Liu S, Liu G, Liu Y, He X (2020) β -glucosidase from *Hevea brasiliensis* seeds: Purification, homology modeling, and insights into the substrate-binding model. *J Food Biochem* **44**: e13206
- Xu S, Zhao C, Wen G, Zhang H, Zeng X, Liu Z, Xu S, Lin C (2018) Longitudinal expression patterns of HMGR, FPS, SS, SE and DS and their correlations with saponin contents in green-purple transitional aerial stems of *Panax notoginseng*. *Ind Crop Prod* **119**: 132 - 143
- Zhang R, Tan SQ, Zhang BL, Guo ZY, Tian LY, Weng P, Luo ZY (2021) Two key amino acids variant of α -l-arabinofuranosidase from *Bacillus subtilis* Str. 168 with altered activity for selective conversion ginsenoside Rc to Rd. *Molecules* **26**: 1733
- Zhu L, Wang Y, Zhao J, Wen M, Li M, Han X (2020) Transformation of Ginsenoside Rb-3 and C-Mx by Recombinant beta-Xylosidase. *Chem J Chin U* **41**: 1010 - 1017

REVIEWERS' COMMENTS

Reviewer #1 (Remarks to the Author):

The authors have responded to reviewers' concern and the manuscript is now acceptable for publication in Nature Communications.

Reviewer #2 (Remarks to the Author):

The paper has been much approved and the authors have responded very carefully to all comments and suggestions from the reviewers.

Reviewer #3 (Remarks to the Author):

The authors have addressed my questions very well. I haven't any further comments.